# T-LoHo: A Bayesian Regularization Model for Structured Sparsity and Smoothness on Graphs

**Changwoo J. Lee**
Department of Statistics
Texas A&M University
c.lee@stat.tamu.edu

**Zhao Tang Luo**
Department of Statistics
Texas A&M University
ztluo@stat.tamu.edu

**Huiyan Sang**
Department of Statistics
Texas A&M University
huiyan@stat.tamu.edu

## Abstract

Graphs have been commonly used to represent complex data structures. In models dealing with graph-structured data, multivariate parameters may not only exhibit sparse patterns but have structured sparsity and smoothness in the sense that both zero and non-zero parameters tend to cluster together. We propose a new prior for high-dimensional parameters with graphical relations, referred to as the Tree-based Low-rank Horseshoe (T-LoHo) model, that generalizes the popular univariate Bayesian horseshoe shrinkage prior to the multivariate setting to detect structured sparsity and smoothness simultaneously. The T-LoHo prior can be embedded in many high-dimensional hierarchical models. To illustrate its utility, we apply it to regularize a Bayesian high-dimensional regression problem where the regression coefficients are linked by a graph, so that the resulting clusters have flexible shapes and satisfy the cluster contiguity constraint with respect to the graph. We design an efficient Markov chain Monte Carlo algorithm that delivers full Bayesian inference with uncertainty measures for model parameters such as the number of clusters. We offer theoretical investigations of the clustering effects and posterior concentration results. Finally, we illustrate the performance of the model with simulation studies and a real data application for anomaly detection on a road network. The results indicate substantial improvements over other competing methods such as the sparse fused lasso. Code is available at https://github.com/changwoo-lee/TLOHO.

## 1 Introduction

In high-dimensional models such as linear regressions where the number of parameters $p$ may exceed the sample size $n$, it is often assumed that the parameter vector $\boldsymbol{\beta} \in \mathbb{R}^p$ has many zero components, namely the *sparsity assumption*. This sparsity assumption allows the unknown parameter $\boldsymbol{\beta}$ to lie on a low-dimensional subspace of $\mathbb{R}^p$, which addresses overfitting and leads to improved predictions and easier interpretations (Hastie et al., 2015). In many real-life applications, however, the parameter of interest $\boldsymbol{\beta}$ should be understood in a certain context where specific data structures exist, such as in time series, spatial, image, and network data analysis. In many such cases, it is desirable to consider another type of low-dimensional structure, where $\boldsymbol{\beta}$ is assumed to have clustered patterns and many possibly clustered zeros, which we shall call *sparse homogeneity assumption*.

One of the most popular models which assumes sparse homogeneity is the sparse fused lasso (Tibshirani et al., 2005) in linear regression settings. 1-D sparse fused lasso imposes an $\ell_1$ penalty on the differences of time-neighboring coefficients as well as individual coefficients. The notion of 'time-neighboring' can be further generalized by considering a graph $G = (V, E)$ with $|V| = p$ so that $G$ represents the general context or structure in which $\boldsymbol{\beta}$ should be interpreted. It leads to the generalized lasso (Tibshirani et al., 2011) where the penalty term now involves $m := |E|$ number of pairwise differences of neighboring vertices with respect to $G$ which facilitates $\beta_i = \beta_j$, where

35th Conference on Neural Information Processing Systems (NeurIPS 2021).

$(i, j) \in E$. In other words, the generalized lasso now allows neighboring coefficients with respect to the graph $G$ to be clustered together. Other similar graph-based regularization approaches include graph OSCAR (Yang et al., 2012), grouping pursuit (Zhu et al., 2013) and graph trend filtering (Wang et al., 2016). However, when the number of edges, $m$, is large relative to $p$, regularization over the whole graph structure often faces severe computational burden. Ke et al. (2015) and Tang and Song (2016) addressed this problem by constructing a preliminary ranking of coefficients and performing a segmentation using 1-D fused lasso. Padilla et al. (2017) used depth-first search (DFS) algorithm to establish a chain graph order for constructing generalized lasso. In a spatial setting, Li and Sang (2019) used a Euclidean minimum spanning tree to form the fused lasso penalty which also enjoys computational benefits. Nevertheless, these fixed chain or tree orders may not be compatible with the true cluster with respect to its context $G$ and hence may lead to over-clustering. Besides, these optimization-based penalized estimators do not usually come with uncertainty measures.

Bayesian regularization methods have gained great popularity in high-dimensional models due to its flexibility and convenience in quantifying estimation and prediction uncertainties. There is rich literature in Bayesian high-dimensional regression models under the sparsity assumption, which can be roughly summarized into two categories: (i) spike-and-slab priors (George and McCulloch, 1993, 1997) and (ii) global-local shrinkage priors (see Bhadra et al., 2019, and references therein). A particular shrinkage prior, called the horseshoe prior (Carvalho et al., 2010), has gained a lot of attraction due to its tail-robustness and super-efficiency as well as substantial computational benefits of the global-local shrinkage prior family (Polson and Scott, 2010). However, relatively few works have been done in the area of Bayesian high-dimensional regression models under the sparse homogeneity assumption. Most approaches put sparsity-inducing priors directly on the pairwise differences of neighboring vertices of a graph to achieve the graph-structured homogeneity of $\boldsymbol{\beta}$ (Kyung et al., 2010; Shimamura et al., 2019; Song and Cheng, 2020; Kim and Gao, 2020; Banerjee, 2021). But similar to the generalized lasso, unless resorting to approximate methods such as EM algorithm, these approaches become computationally expensive to get posterior samples for general graphs with a large number of edges. Another more subtle limitation of these methods is that they fail to incorporate structural assumptions among local shrinkage parameters.

In this paper, we propose a Bayesian Tree-based Low-rank Horseshoe (T-LoHo) model to identify structured sparsity and smoothness of parameters whose prior structure is represented by a graph. Unlike other existing Bayesian methods which put independent priors on pairwise differences, T-LoHo extends the univariate horseshoe shrinkage prior to a multivariate setting where $\boldsymbol{\beta}$ and its local shrinkage parameters are assumed to be piecewise constants on a graph. This low-rank structured model for local shrinkage parameters allows both clustering and sparsity effects to have strong local adaptivities. A random spanning forest (RSF) based graph clustering prior is introduced to adaptively learn a compatible neighboring order to model graph partitions, which extends the recently developed random spanning tree partition models (Teixeira et al., 2019; Luo et al., 2021) to allow for possibly unconnected graphs. We show that this RSF-based prior retains model richness as its support is flexible enough to accommodate all possible contiguous partitions. We introduce T-LoHo prior to model high-dimensional linear regression coefficients, although it can be flexibly embedded in other high-dimensional models. The resulting cluster estimates not only provide uncertainty measures but also have strong flexibility to accommodate sharp discontinuities. With state of the art computational strategies, we provide a highly efficient MCMC algorithm for posterior inference utilizing tree structures. We also study theoretical aspects of our model including posterior consistency results and its effect on clustering. We demonstrate the efficacy of T-LoHo model by using synthetic data and a real data analysis for anomaly detection on the Manhattan road network.

## 2 T-LoHo: Tree-based Low-rank Horseshoe Model

Consider a graph $G = (V, E)$ with $|V| = p$ and $|E| = m$ which represents the pre-known structure of our parameter of interest $\boldsymbol{\beta} \in \mathbb{R}^p$. For example, when $\boldsymbol{\beta}$ has a certain order (e.g. time), then we set $G$ as a linear chain graph. When (vectorized) $\boldsymbol{\beta}$ lies on a 2-D image, we let $G$ be a 2-D lattice. A *contiguous graph partition* $\Pi = \{\mathcal{C}_1, \ldots, \mathcal{C}_K\}$ is a disjoint partition of $V$ such that each $\mathcal{C}_k$ induces a connected subgraph of $G$ (the term *graph partition* will always refer to a contiguous graph partition hereafter). Under the sparse homogeneity assumption on $\boldsymbol{\beta}$, the goal is to find a graph partition $\Pi$ with unknown size $K$ and corresponding estimate of $\boldsymbol{\beta}$ which contains many zeros. We introduce the model for $\boldsymbol{\beta}$ conditional on a partition $\Pi$ in Section 2.1 and describe the model for $\Pi$ in Section 2.2.

## 2.1 Low-Rank Multivariate Horseshoe Prior

First we review the horseshoe prior (Carvalho et al., 2010), which provides a sparse estimate of $\boldsymbol{\beta}$ by shrinking small effects towards zero while maintaining robustness to large signals:

$$\boldsymbol{\beta}|\sigma^2, \tau^2, \{\lambda_j\}_{j=1}^p \sim \mathcal{N}_p(\mathbf{0}, \sigma^2\tau^2 \mathrm{diag}(\lambda_1^2, \ldots, \lambda_p^2)),$$

$$\lambda_j \overset{iid}{\sim} C^+(0,1), \quad \tau \sim C^+(0, \tau_0), \quad p(\sigma^2) \propto 1/\sigma^2,$$

where $\mathcal{N}_p(\boldsymbol{m}, \boldsymbol{V})$ denotes a $p$-dimensional multivariate normal distribution with mean $\boldsymbol{m}$ and covariance $\boldsymbol{V}$, and $C^+(0, \gamma)$ denotes a half-Cauchy distribution with density $p(\lambda) = 2/[\pi\gamma(1 + \lambda^2/\gamma^2)]$ for $\lambda > 0$. Here $\tau$ is a global shrinkage parameter with hyperparameter $\tau_0$ to enforce global shrinkage towards zero, $\{\lambda_j\}$ are local shrinkage parameters with heavy tails which allow some of the $\beta_j$'s to escape the shrinkage, and $\sigma^2$ is a scaling factor which is often assumed to be the noise variance.

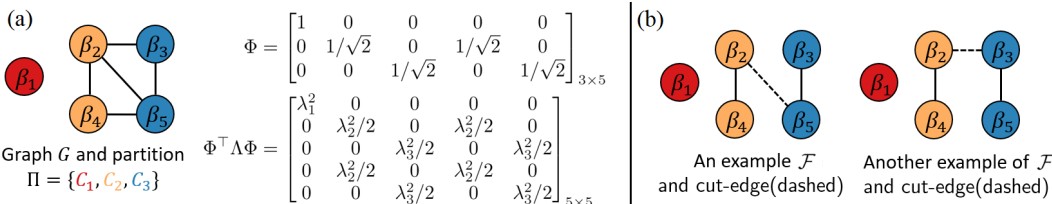

Figure 1: Illustrative example of (a) graph partition and corresponding parameters when $\boldsymbol{\beta} \in \mathbb{R}^5$ forms $K = 3$ clusters, $\mathcal{C}_1 = \{1\}, \mathcal{C}_2 = \{2, 4\}, \mathcal{C}_3 = \{3, 5\}$; (b) compatible forest $\mathcal{F}$ and cut-edge(s).

Given a graph partition $\Pi$ of $G$, we seek a new joint prior distribution of $\boldsymbol{\beta} \in \mathbb{R}^p$ to impose sparse homogeneity assumption while incorporating the clustering structure of $\Pi$. For example when $p = 5$ and $\Pi = \{\{1\}, \{2, 4\}, \{3, 5\}\}$ as shown in Figure 1(a), we seek a distribution which lies on a 3-dimensional subspace $\{\boldsymbol{\beta} \in \mathbb{R}^5 : \beta_2 = \beta_4, \beta_3 = \beta_5\}$. In general, given a graph partition $\Pi$ with size $K$, we can construct a $K \times p$ matrix $\Phi$ from $\Pi$:

$$\Phi_{kj} = 1/\sqrt{|\mathcal{C}_k|} \text{ if } j \in \mathcal{C}_k \text{ and } 0 \text{ otherwise}, \quad k = 1, \ldots, K, \ j = 1, \ldots, p. \tag{1}$$

This $K \times p$ matrix $\Phi$ represents a projection from a $p$-dimensional space to a reduced $K$-dimensional space. Note that rows of $\Phi$ are mutually orthonormal so that $\Phi\Phi^\top = \mathbf{I}_K$. Using $\Phi$ obtained from $\Pi$, we introduce the low-rank horseshoe (LoHo) prior to impose sparse homogeneity assumption on $\boldsymbol{\beta}$:

$$\boldsymbol{\beta} \,|\, \sigma^2, \tau^2, \Lambda, \Pi \sim \mathcal{N}_p(\mathbf{0}, \sigma^2\tau^2\Phi^\top\Lambda\Phi), \ \Lambda := \mathrm{diag}(\lambda_1^2, \ldots, \lambda_K^2), \tag{2}$$

$$\lambda_k \overset{iid}{\sim} C^+(0,1), \quad \tau \sim C^+(0, \tau_0), \quad p(\sigma^2) \propto 1/\sigma^2. \tag{3}$$

See fig. 1(a) for an example of $\Phi$ and a covariance matrix $\Phi^\top\Lambda\Phi$. Note LoHo introduces a covariance matrix capturing clustering dependence among $\boldsymbol{\beta}$ and its marginal local shrinkage for simultaneous sparsity and fusion, a key distinction from most existing methods (Carvalho et al., 2010; Kyung et al., 2010; Shimamura et al., 2019) that assume independence or only dependence among $\boldsymbol{\beta}$ but not local shrinkage parameters. Since $K$ is assumed to be small relative to $p$, the covariance matrix $\Phi^\top\Lambda\Phi$ has low rank (i.e., $\mathrm{rank}(\Phi^\top\Lambda\Phi) = K \ll p$), and thus LoHo does not have a density with respect to Lebesgue measure on $\mathbb{R}^p$. Instead, its distribution lies on the row space of $\Phi$ with $\dim(\mathrm{rowsp}(\Phi)) = K$, and we can consider the transformation $\tilde{\boldsymbol{\beta}} = \Phi\boldsymbol{\beta}$ so that $\tilde{\boldsymbol{\beta}}$ has a $\mathcal{N}_K(\mathbf{0}, \sigma^2\tau^2\Lambda)$ density with respect to Lebesgue measure on $\mathrm{rowsp}(\Phi)$. Observe that $\Phi^\top$ is the Moore–Penrose pseudoinverse of $\Phi$, which implies $\Phi^\top\Phi$ is a projection onto $\mathrm{rowsp}(\Phi)$ so that we can recover $\boldsymbol{\beta} = \Phi^\top\tilde{\boldsymbol{\beta}}$.

By assigning a half-Cauchy prior, global shrinkage parameter $\tau$ creates a strong pull towards zero while *clusterwise* local shrinkage parameters $\{\lambda_k\}_{k=1}^K$ allow some of the *clusterwise* $\tilde{\beta}_k$'s to escape the shrinkage. Assuming $\Phi$ having orthonormal rows is important in the sense that the effect of local shrinkage parameters $\{\lambda_k\}_{k=1}^K$ remains same across the clusters with varying size. Thus using the projection onto the low-dimensional subspace, LoHo gives more parsimonious estimate of $\boldsymbol{\beta}$ by forming clusters of zero and non-zero parameters.

LoHo can be naturally incorporated into a linear model. With response vector $\boldsymbol{y} \in \mathbb{R}^n$ and column-standardized design matrix $\mathbf{X} \in \mathbb{R}^{n \times p}$ so that each column has unit $\ell_2$ norm, we can write

$$\boldsymbol{y} = \mathbf{X}\boldsymbol{\beta} + \boldsymbol{\epsilon}, \qquad \boldsymbol{\epsilon} \sim \mathcal{N}_n(\mathbf{0}, \sigma^2\mathbf{I}_n)$$

Under this formulation, LoHo has a close connection with Bayesian compressed regression (Guhaniyogi and Dunson, 2015). It randomly projects predictors $\mathbf{X}_i \mapsto \Phi \mathbf{X}_i$ with a certain family of matrix $\Phi$ and performs model averaging, at the cost of losing interpretability of $\boldsymbol{\beta}$. But LoHo directly introduces a prior on $\boldsymbol{\beta}$ using the projection matrix $\Phi$ defined as (1) so that it induces clustered coefficient while maintaining interpretability of $\boldsymbol{\beta}$. Also, functional horseshoe (Shin et al., 2020) shrinks $\mathbf{X}\boldsymbol{\beta}$ towards the subspace of $\text{colsp}(\mathbf{X})$ while LoHo shrinks $\boldsymbol{\beta}$ towards $\mathbf{0}$ along the $\text{rowsp}(\Phi)$.

We remark that, although it appears $(\tau, \Lambda)$ and $\Phi$ handle sparsity and homogeneity separately, shrinkage component in LoHo also affects clustering by facilitating cluster fusion when signal is low and improving cluster identification when signal is high. More details are discussed in Section 3.2.

## 2.2 Tree-based Graph Partition Prior

Now we describe how we model the unknown partition $\Pi$. A graph partition can be defined as a collection of disjoint connected subgraphs such that the union of vertices is $V$. To bypass the need to handle a complex combinatorial graph partition problem, we consider an equivalent formulation of graph partition through edge cuts of spanning forests of $G$. Prop. 1 guarantees that for *any* choice of partition $\Pi$, there exist a spanning forest $\mathcal{F}$ and a corresponding set of cut-edges $E^C$ that *induce* $\Pi$, i.e., some edges in $\mathcal{F}$ can be removed so that vertices connected to each other form a cluster.

**Proposition 1.** *Let $G = (V, E)$ be a graph with $n_c$ connected components and $\Pi = \{\mathcal{C}_1, \ldots, \mathcal{C}_K\}$ be a graph partition of $G$. Then there exists a spanning forest $\mathcal{F} = (V, E^F)$ with $|E^F| = |V| - n_c$, and a set of cut-edges $E^C \subset E^F$ with $|E^C| = K - n_c$ such that $\mathcal{F}$ and $E^C$ induce $\Pi$.*

Proof is deferred to Appendix A2. We will say a spanning forest $\mathcal{F}$ is *compatible* with $\Pi$ if we can construct $\Pi$ by cutting some of its edges. See Figure 1(b) for two examples of compatible $\mathcal{F}$ and cut-edge(s). Prop.1 suggests that to induce a graph partition prior with full support, it amounts to first assigning a prior model on all possible spanning forests of $G$, and then assigning a prior model on all possible cut-edges sets conditional on a spanning forest. When $G$ is a connected spatial or spatial temporal graph, Teixeira et al. (2019) considered a discrete uniform prior over all possible spanning trees. But in this case, approximate method is required to sample a spanning tree from its full conditional distribution due to serious inefficiency of the rejection sampler. In contrast, Luo et al. (2021) considered a random minimum spanning tree approach by assigning iid uniform random weights to the edges, so it can generate any arbitrarily given spanning tree of $G$ and enable an exact and efficient posterior conditional sampling algorithm. Thus we follow a similar approach as in Luo et al. (2021) which leads to the following random minimum spanning forest prior on $\mathcal{F}$ with full support,

$$\mathcal{F} = \text{MSF}(G, \boldsymbol{W}), \quad W_{ij} \overset{iid}{\sim} \text{Unif}(0,1), (i,j) \in E, \tag{4}$$

where $\text{MSF}(G, \boldsymbol{W})$ denotes the minimum spanning forest of the graph $G$ with edge weights $\boldsymbol{W}$.

After $\mathcal{F}$ is given, selecting $K - n_c$ cut-edges forms a partition $\Pi$ with size $K$. Following approaches of Knorr-Held and Raßer (2000), Feng et al. (2016), and Luo et al. (2021), we introduce a geometrically decaying prior distribution on $K$, and then select cut-edges uniformly at random given $(\mathcal{F}, K)$. The following prior specification completes the T-LoHo model:

$$\Pr(K = k) \propto (1 - c)^k, \quad k = n_c, n_c + 1, \cdots, p, \quad c \in [0, 1) \tag{5}$$

$$p(\Pi \mid \mathcal{F}, K) \propto 1(\mathcal{F} \text{ is compatible with } \Pi \text{ and } |\Pi| = K). \tag{6}$$

T-LoHo involves two hyperparameters related to model complexity penalization. One is $c$ in (5) which, if selected to be closer to 1, strongly penalizes models with larger numbers of clusters. Another is $\tau_0$ in (3) controlling the strength of global shrinkage. As $\tau_0$ reduces to 0, the posterior distribution of $\tilde{\boldsymbol{\beta}}$ tends to concentrate more at zero. More detailed hyperparameter sensitivity analysis and selection criteria are deferred to Appendix A5.

# 3 Posterior Inference and Theoretical Properties

## 3.1 Posterior Sampler and Computational Strategies

Here we briefly describe a reversible-jump Markov chain Monte Carlo algorithm (RJMCMC) (Green, 1995) and discuss computational strategies therein. Denote $\Theta := (\tilde{\boldsymbol{\beta}}, \sigma^2, \Lambda, \tau, \Pi, K, \mathcal{F})$ be the set of parameters, and also denote $\tilde{\mathbf{X}} := \mathbf{X}\Phi^\top$ so that $\tilde{\mathbf{X}}\tilde{\boldsymbol{\beta}} = \mathbf{X}\Phi^\top\Phi\boldsymbol{\beta} = \mathbf{X}\boldsymbol{\beta}$ since $\boldsymbol{\beta} \in \text{rowsp}(\Phi)$.

The posterior $p(\Theta|\boldsymbol{y})$ is

$$p(\Theta|\boldsymbol{y}) \propto \quad \mathcal{N}_n(\boldsymbol{y}|\tilde{\mathbf{X}}\tilde{\boldsymbol{\beta}}, \sigma^2\mathbf{I}_n) \times \mathcal{N}_K(\tilde{\boldsymbol{\beta}}|\mathbf{0}, \sigma^2\tau^2\Lambda) \times 1/\sigma^2 \tag{7}$$

$$\times (1+\tau^2)^{-1} \prod_{k=1}^K (1+\lambda_k^2)^{-1} \times \binom{p-n_c}{K-n_c}^{-1} \times (1-c)^K \times 1 \tag{8}$$

where line (8) is the product of priors $p(\tau)\prod_{k=1}^K p(\lambda_k)p(\Pi|K,\mathcal{F})p(K)p(\boldsymbol{W})$. We draw posterior samples of $\Theta|\boldsymbol{y}$ using a collapsed RJMCMC posterior sampler as described in Algorithm 1.

---

**Algorithm 1:** One full iteration of the RJMCMC posterior sampler

---

**Step 1.** Update $\Pi, K, \mathcal{F}$ using collapsed conditional $[\Pi, K, \mathcal{F}|\Lambda, \tau, \boldsymbol{y}]$ where $\tilde{\boldsymbol{\beta}}, \sigma^2$ are integrated out.[†] With probabilities $(p_a, p_b, p_c, p_d)$ summing up to 1, perform one of the following substeps:

    1-a. (*split*) Propose $(\Pi^\star, K^\star = K+1)$ compatible with $\mathcal{F}$, and accept with probability $\min\{1, \mathcal{A}_a \cdot \mathcal{P}_a \cdot \mathcal{L}_a\}$, where $\mathcal{A}_a$ is prior ratio, $\mathcal{P}_a$ is proposal ratio, $\mathcal{L}_a$ is likelihood ratio.

    1-b. (*merge*) Propose $(\Pi^\star, K^\star = K-1)$ compatible with $\mathcal{F}$, and accept w.p. $\min\{1, \mathcal{A}_b \cdot \mathcal{P}_b \cdot \mathcal{L}_b\}$.

    1-c. (*change*) Propose $(\Pi^\star, K^\star = K)$ compatible with $\mathcal{F}$, and accept w.p. $\min\{1, \mathcal{A}_c \cdot \mathcal{P}_c \cdot \mathcal{L}_c\}$.

    1-d. (*hyper*) Update $\mathcal{F}$ compatible with current $\Pi$.

**Step 2.** Jointly update $(\tau, \sigma^2, \tilde{\boldsymbol{\beta}})$ from $[\tau, \sigma^2, \tilde{\boldsymbol{\beta}} \,|\, \Lambda, \Pi, K, \mathcal{F}, \boldsymbol{y}]$, by performing:

    2-1. Update $\tau$ from $[\tau \,|\, \Lambda, \Pi, K, \mathcal{F}, \boldsymbol{y}]$ using a Metropolis-Hastings sampler,

    2-2. Update $\sigma^2$ from $[\sigma^2 \,|\, \tau, \Lambda, \Pi, K, \mathcal{F}, \boldsymbol{y}]$ with an inverse gamma distribution,

    2-3. Update $\tilde{\boldsymbol{\beta}}$ from $[\tilde{\boldsymbol{\beta}} \,|\, \sigma^2, \tau, \Lambda, \Pi, K, \mathcal{F}, \boldsymbol{y}]$ with a multivariate normal distribution.

**Step 3.** Update $\Lambda$ from $[\Lambda \,|\, \tau, \sigma^2, \tilde{\boldsymbol{\beta}}, \Pi, K, \mathcal{F}, \boldsymbol{y}]$ using a slice sampler.

---

[†] When $\mathbf{X} = \mathbf{I}_n$ (i.e., normal means model), it is possible to integrate out $\Lambda$ instead of $\sigma^2$; see Appendix A1.

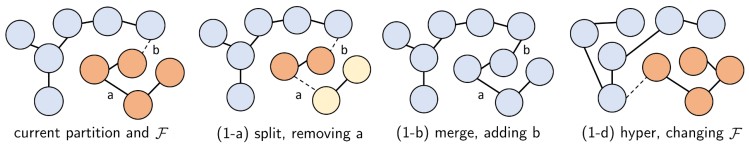

current partition and $\mathcal{F}$     (1-a) split, removing a     (1-b) merge, adding b     (1-d) hyper, changing $\mathcal{F}$

Figure 2: Illustration of step 1 in Alg.1. Step (1-c) corresponds to performing (1-a), (1-b) sequentially.

Step 1 updates cluster assignment by proposing one of the four moves (fig. 2); see Appendix A1 for details on acceptance probabilities. Here instead of using the full conditional distribution, we use the collapsed conditional distribution $[\Pi, K, \mathcal{F}|\Lambda, \tau, \boldsymbol{y}]$ where $(\boldsymbol{\beta}, \sigma^2)$ are integrated out to calculate the likelihood ratio $\mathcal{L}$ which significantly improves mixing. Specifically, the collapsed conditional $[\Pi, K, \mathcal{F}|\Lambda, \tau, \boldsymbol{y}]$ is proportional to $|\boldsymbol{\Sigma}|^{-1/2}(\boldsymbol{y}^\top\boldsymbol{\Sigma}^{-1}\boldsymbol{y}/2)^{-n/2}$ where $\boldsymbol{\Sigma} = \mathbf{I}_n + \tau^2\tilde{\mathbf{X}}\Lambda\tilde{\mathbf{X}}^\top$ so that

$$\text{Likelihood ratio } \mathcal{L} = \frac{|\boldsymbol{\Sigma}^\star|^{-1/2}(\boldsymbol{y}^\top\boldsymbol{\Sigma}^{\star-1}\boldsymbol{y}/2)^{-n/2}}{|\boldsymbol{\Sigma}|^{-1/2}(\boldsymbol{y}^\top\boldsymbol{\Sigma}^{-1}\boldsymbol{y}/2)^{-n/2}}. \tag{9}$$

Here superscript $^\star$ indicates the proposed parameters. Step 2 jointly updates $[\tau, \sigma^2, \tilde{\boldsymbol{\beta}}|-] = [\tau|-] \times [\sigma^2|\tau, -] \times [\tilde{\boldsymbol{\beta}}|\tau, \sigma^2, -]$ following the approach of Johndrow et al. (2020). Finally, step 3 updates local shrinkage parameters $\Lambda$ using the slice sampler (Neal, 2003; Polson et al., 2014).

One demanding computational bottleneck is the likelihood ratio calculation that involves $\boldsymbol{\Sigma}^{-1}$ and $|\boldsymbol{\Sigma}|$. A notable advantage of T-LoHo is that it projects $\mathbf{X}$ to a low-dimensional space under the sparse homogeneity assumption, leading to an $n$ by $K$ transformed design matrix $\tilde{\mathbf{X}} = \mathbf{X}\Phi^\top$ where typically $K \ll n$. Applying Sherman-Woodbury-Morrison formula, we can easily calculate

$$\boldsymbol{\Sigma}^{-1} = (\mathbf{I}_n + \tau^2\tilde{\mathbf{X}}\Lambda\tilde{\mathbf{X}}^\top)^{-1} = \mathbf{I}_n - \tilde{\mathbf{X}}(\tau^{-2}\Lambda^{-1} + \tilde{\mathbf{X}}^\top\tilde{\mathbf{X}})^{-1}\tilde{\mathbf{X}}^\top$$

by reducing the rank of the inverting matrix from $n$ to $K$. Furthermore, we utilize Cholesky decomposition $\mathbf{R}^\top\mathbf{R} = \tau^{-2}\Lambda^{-1} + \tilde{\mathbf{X}}^\top\tilde{\mathbf{X}}$ to efficiently calculate $\boldsymbol{\Sigma}^{-1}$ and $|\boldsymbol{\Sigma}|$. Once we have right triangular $\mathbf{R}$, calculation of $\boldsymbol{\Sigma}^{-1}$ only involves backward substitution, and also we can calculate $|\boldsymbol{\Sigma}|$ at no extra cost using the matrix determinant lemma. A naive implementation of the above likelihood

calculation would require to repeatedly update $\tau^{-2}\Lambda^{-1} + \tilde{\mathbf{X}}^\top\tilde{\mathbf{X}}$ and its Cholesky factor $\mathbf{R}$ for each MCMC step, except in steps (2-2) and (2-3) where we use $\mathbf{\Sigma}^{-1}$ and $\mathbf{R}$ calculated from step (2-1) respectively via Rue (2001)'s algorithm. For instance, step 1 has a new $\tilde{\mathbf{X}}^\star = \mathbf{X}\Phi^{\star\top}$ and steps (2-1) and 3 have a new $\tau^\star$ or $\Lambda^\star$. Alternatively, we propose to directly update $\mathbf{R}$ from its previous value. For step 1 where $\tilde{\mathbf{X}}^\top\tilde{\mathbf{X}}$ changes, we use Cholesky rank-1 update/downdate (Golub and Van Loan, 2013, Sec. 6.5.4) which reduces computational cost from $O\big(\max\{nK^2, K^3\}\big)$ to $O(K^2)$. For steps (2-1) and 3 where the diagonal part changes, we present a direct algorithm (see Alg. 2 in Appendix A1) for updating the corresponding Cholesky factor. These Cholesky updating schemes prove to be simple yet powerful computational strategies as MCMC typically requires many iterations.

Excluding step (1-d), the total computational cost of Alg. 1 is $O\big(\max\{nK, K^3\}\big)$, compared to the direct computation of $\mathbf{\Sigma}^{-1}$ and $|\mathbf{\Sigma}|$ which leads to $O(n^3)$. Step (1-d) takes $O(m\log p)$ to construct MSF where $m$ is the number of edges in $G$. Note that step 1-d (hyper) is selected with probability $p_d$ at each iteration. We suggest a small value of $p_d$ such as $0.05$, so that it would save computation time and give RJMCMC enough iterations to explore the partition compatible with the current MSF $\mathcal{F}$.

### 3.2 Clustering Effect of T-LoHo

In this section, we investigate how T-LoHo prior differs from the Gaussian prior in terms of the effect on clustering. We focus on the sparsity of *edge differences* because of its important role in clustering. Examples include $\ell_1$ (Tibshirani et al., 2011) and $\ell_0$ penalties (Fan and Guan, 2018) on $\{\beta_i - \beta_j : (i,j) \in E\}$. Existing Bayesian methods seek sparsity of $\beta_i - \beta_j$ by putting suitable prior distributions on edge differences, such as Laplace (Kyung et al., 2010), normal-exponential-gamma (Shimamura et al., 2019), student's t (Song and Cheng, 2020; Banerjee and Shen, 2020), spike-and-slab (Kim and Gao, 2020), and horseshoe (Banerjee, 2021). But the aforementioned Bayesian methods have several limitations: (i) an additional post-processing step is often required due to the lack of explicit cluster estimates; (ii) sparsity assumption cannot be easily incorporated into their methods; and (iii) posterior inference method is either inefficient or inflexible (e.g., relying on a single $\mathcal{F}$) when the underlying graph $G$ has many edges. In contrast, although T-LoHo does not put sparsity-inducing priors directly on edge differences, it overcomes all these limitations and effectively finds clusters by its flexible low-rank structure. Below, we show that the use of horseshoe prior not only introduces shrinkage of $\boldsymbol{\beta}$ but also has a less obvious but profound clustering effect to facilitate homogeneity pursuit.

To analyze the effect of T-LoHo prior on clustering, we focus on the simple case when $\mathbf{X} = \mathbf{I}_n$ so that observations $y_i$ are independent and normally distributed, i.e., $y_i|\beta_i, \sigma^2 \sim \mathcal{N}(\beta_i, \sigma^2)$, $i = 1, \ldots, n$. Without loss of generality, consider the merge step (1-b) where the comparison between the proposed merged model $\mathcal{M}_1 := (\Pi^\star, K-1)$ by combining two existing clusters $\mathcal{C}_1, \mathcal{C}_2$ and the current model $\mathcal{M}_2 := (\Pi, K)$ is made:
$$\mathcal{M}_1 : \text{mean of } \{y_i\}_{i\in\mathcal{C}_1} = \mu_1 = \mu_2 = \text{mean of } \{y_i\}_{i\in\mathcal{C}_2} \quad \text{v.s.} \quad \mathcal{M}_2 : \mu_1 \neq \mu_2$$
We analyze the acceptance probability $\min\{1, \mathcal{A}\cdot\mathcal{P}\cdot\mathcal{L}\}$ in step (1-b) of Alg. 1 because it is crucial in the clustering mechanism. The term $\mathcal{A}\cdot\mathcal{P}$ is nothing but $1/(1-c)$ which reflects the model size penalty imposed by $p(K)$. The key part is the likelihood ratio $\mathcal{L}$, where the different choice of prior on $\boldsymbol{\beta}$ (equivalently cluster mean $\mu$) leads to the different $\mathcal{L}$ under the same data. This likelihood ratio corresponds to the Bayes factor (Kass and Raftery, 1995) of the Bayesian two-sample t test (Gönen et al., 2005). Thus, here we compare the Bayes factor $\text{BF}_{12} = p(\text{data}|\mathcal{M}_1)/p(\text{data}|\mathcal{M}_2)$ under the normal and T-LoHo prior respectively to analyze their effects on clustering.

Following the formulation of Bayesian two-sample t test, priors are reparametrized as $(\delta, \bar{\mu}, \sigma^2) := ((\mu_1 - \mu_2)/\sigma, (\mu_1 + \mu_2)/2, \sigma^2)$ where the standardized difference $\delta = (\mu_1 - \mu_2)/\sigma$ is the parameter of interest. Here we assume a noninformative prior on nuisance parameters $p(\bar{\mu}, \sigma^2) \propto 1/\sigma^2$, since otherwise Bayes factor is no longer a function of two-sample t statistics $t$ and $(n_1, n_2)$:
$$t = (\bar{y}_1 - \bar{y}_2)/(s_p/\sqrt{n_\delta}), \quad s_p^2 = \big((n_1-1)s_1^2 + (n_2-1)s_2^2\big)/\nu, \quad n_\delta = (n_1^{-1} + n_2^{-1})^{-1}$$
where $n_k = |\mathcal{C}_k|$, $\nu = n_1 + n_2 - 2$, and $\bar{y}_k, s_k^2$ are the sample mean and variance of group $k$, $k = 1, 2$.

It is obvious that independent normal priors on $\mu_k$ (with variance scaled with $\sigma^2$) leads to a normal prior on $\delta$ as well. When $\delta \sim \mathcal{N}(0,1)$, the Bayes factor $\text{BF}_{12}^n$ (Gönen et al., 2005) is
$$\text{BF}_{12}^n = \frac{\int p(\text{data}|\delta=0, \bar{\mu}, \sigma^2)p(\bar{\mu}, \sigma^2)d(\bar{\mu}, \sigma^2)}{\int p(\text{data}|\delta, \bar{\mu}, \sigma^2)p(\delta, \bar{\mu}, \sigma^2)d(\delta, \bar{\mu}, \sigma^2)} = \frac{\big(1 + t^2/\nu\big)^{-(\nu+1)/2}}{(1 + n_\delta)^{-1/2}\big\{1 + t^2/\left[\nu(1+n_\delta)\right]\big\}^{-(\nu+1)/2}}.$$

Now if we change the priors on $\mu_1$ and $\mu_2$ from independent normal to independent horseshoe distributions, it induces the heavy-tailed prior $\pi_\Delta$ on $\delta$ which is a convolution of two horseshoe priors:

**Proposition 2.** *Let $\pi_{HS}(\mu|\sigma,\tau) = \int_0^\infty \mathcal{N}(\mu|0,\sigma^2\tau^2\lambda^2)C^+(\lambda|0,1)d\lambda$ be a horseshoe prior. If $\mu_1 \sim \pi_{HS}(\sigma,\tau_1)$ and $\mu_2 \sim \pi_{HS}(\sigma,\tau_2)$ independently, then it induces a distribution of the standardized difference $\delta := (\mu_1 - \mu_2)/\sigma$ given $\tau_1,\tau_2$, denoted as $\pi_\Delta(\delta|\tau_1,\tau_2)$, which can be written as a scale mixture of normal with mixing distribution $f_W(w)$ where $w > 0$:*

$$\pi_\Delta(\delta|\tau_1,\tau_2) = \int_0^\infty \mathcal{N}(\delta|0,w)f_W(w)dw, \quad f_W(w) = \frac{1}{\pi}\frac{\tau_1\sqrt{w+\tau_1^2} + \tau_2\sqrt{w+\tau_2^2}}{\sqrt{w+\tau_1^2}\sqrt{w+\tau_2^2}(w+\tau_1^2+\tau_2^2)}$$

Proof is deferred to Appendix A3. See left panel of fig. 3 for the graphical illustration of $\pi_\Delta(\delta|1,1)$. Its tail behaves similarly with the Strawderman-Berger prior (Strawderman, 1971; Berger et al., 1980). By Proposition 2, the Bayes factor $\mathrm{BF}_{12}^{hs}$ under the prior induced by horseshoe $\delta \sim \pi_\Delta(\delta|\tau_1,\tau_2)$ is

$$\mathrm{BF}_{12}^{hs} = \frac{\left(1+t^2/\nu\right)^{-(\nu+1)/2}}{\int_0^\infty (1+n_\delta w)^{-1/2}\left\{1+t^2/[\nu(1+n_\delta w)]\right\}^{-(\nu+1)/2} f_W(w)dw}.$$

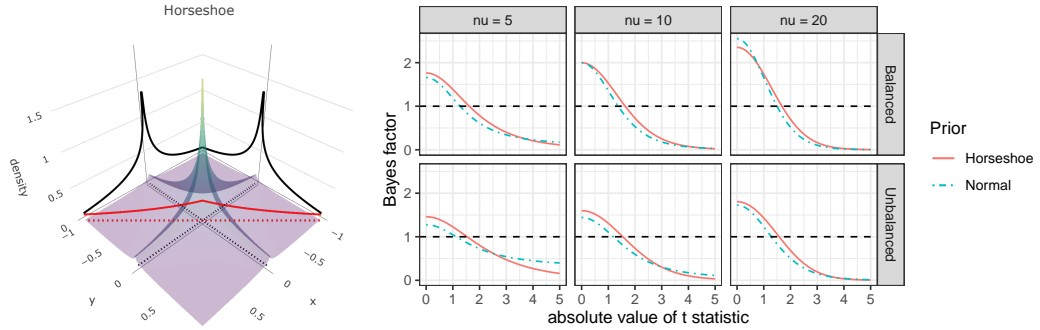

Figure 3: (Left) Joint density $f(x,y) = \pi_{HS}(x)\pi_{HS}(y)$ overlaid with marginal density $\pi_\Delta(x-y)$ shown as red, when $\tau_1 = \tau_2 = 1$. (Right) Comparison of $\mathrm{BF}_{12}$ as a function of $|t|$ under different $(n_1,n_2)$ settings. Higher Bayes factor implies favoring one-group $\mathcal{M}_1 : \mu_1 = \mu_2$.

Under the normal and $\pi_\Delta$ priors on $\delta$, we compare $\mathrm{BF}_{12}$ as a function of $|t|$ when the group sizes are (i) balanced, $n_1 = n_2$; (ii) unbalanced, $n_1 : n_2 = 9 : 1$ with $\nu \in \{5,10,20\}$, and display results at the right panel of fig. 3. Since an arbitrary choice of scale leads to the different BF, we set $\delta \sim \mathcal{N}(0,1)$ which corresponds to the unit information prior (Kass and Wasserman, 1995) and choose the scale of $\delta \sim \pi_\Delta$ such that median of scale mixture distribution $f_W(w)$ becomes 1. See Appendix A3 for details of specific choices of $(\tau_1,\tau_2)$ under different settings of $(n_1,n_2)$. Although this is not the only possible scale matching criterion, it is a reasonable choice for a fair BF comparison so that we can focus on different origin/tail behavior of $\pi_\Delta$ compared to the normal distribution.

When $|t|$ is small to moderate, we can see that $\mathrm{BF}_{12}^{hs}$ is generally greater than $\mathrm{BF}_{12}^n$, except for the case when $n_\delta$ is large (balanced, large $\nu$) and $|t|$ is small (in which case large $\mathrm{BF}_{12}$ in both models leads to accepting merge proposals anyways). This implies the prior $\delta \sim \pi_\Delta$ more strongly favors the one-group $\mathcal{M}_1$ over the two-group $\mathcal{M}_2$ compared to the normal prior $\delta \sim \mathcal{N}(0,1)$. This can be explained by the fact that, under $\mathcal{M}_2$, the heavy-tailed prior $\pi_\Delta$ anticipates a large effect size *a priori* compared to the normal prior. Thus T-LoHo facilitates cluster fusion when $|t|$ is small to moderate.

Now when $|t|$ is large, we can see that $\pi_\Delta$ more strongly favors $\mathcal{M}_2$ compared to the normal prior, and the difference becomes more noticeable when group sizes are small and unbalanced. In fact, when $(n_1,n_2)$ are fixed, $\mathrm{BF}_{12}^{hs}$ converges to 0 as $|t| \to \infty$ whenever $\nu \geq 1$ but $\mathrm{BF}_{12}^n$ never converges to 0 and is lower bounded by $(1+n_\delta)^{-\nu/2} > 0$, which is also known as the *information paradox* (Liang et al., 2008). This finite sample consistency is important since it allows the model to identify the (small, unbalanced) cluster with a high signal difference, which might not be possible under the usual normal prior because the penalty term $p(K) \propto (1-c)^K$ may overwhelm it.

In summary, compared to the normal prior, T-LoHo tends to reduce redundant cluster representations (when $|t|$ is small) while better capturing the highly significant cluster differences (when $|t|$ is large).

### 3.3 Posterior Consistency Results

**Notations.** Let $(\boldsymbol{\beta}^*, \tilde{\boldsymbol{\beta}}^*, \sigma^*)$ denote the true $\boldsymbol{\beta}$, $\tilde{\boldsymbol{\beta}}$ and $\sigma$, respectively. Let $\xi^* = \{j \in V : \beta_j^* \neq 0\}$ denote the true active set of indices. Let $\check{\Pi}$ denote an arbitrary partition of $V = \{1, \cdots, p\}$ whose corresponding partition of $\xi^*$ is determined by removing those edges with $\beta_i^* - \beta_j^* > 0$ from the subgraph of any $\mathcal{F}$ compatible with $\check{\Pi}$ at vertex set $\xi^*$. We define $g_n^* = \max_{\check{\Pi}} |\check{\Pi}(\xi^*)|$ among all possible $\check{\Pi}$. Let $P_n$ denote all unique partitions that have at most $g_n^*(1 + c_\delta)$ clusters and their corresponding partitions of $\xi^*$ are nested in the true partition of $\xi^*$ for some constant $c_\delta > 0$.

Below, we consider the case when $p$ can be much larger than $n$ and establish posterior concentration results for the T-LoHo model as $n$ goes to infinity. Our results rely on the following assumptions:

(A-1) The graph satisfies $g_n^* \prec n/\log p$, $n_c = o(g_n^*)$, and $\log |P_n| = O(g_n^* \log p)$.

(A-2) All the covariates are uniformly bounded. There exists some fixed constant $\lambda_0 > 0$, such that $\lambda_{\min}(\tilde{\mathbf{X}}^T \tilde{\mathbf{X}}) \geq n\lambda_0$ for any partition in $P_n$.

(A-3) $\max_j |\tilde{\beta}_j^*|/\sigma^* < L$, where $\log(L) = O(\log p)$.

(A-4) $-\log \tau = O(\log p)$, $\tau < p^{-(2+c_\tau)}\sqrt{g_n^* \log p/n}$, $1 - c \geq p^{-c_\alpha}$, and $\min_{\sigma^2 \in [\sigma^{*2},\ \sigma^{*2}(1+c_\sigma \varepsilon_n^2)]} \pi(\sigma^2) > 0$ for some positive constants $c_\tau$, $c_\alpha$ and $c_\sigma$.

Assumption (A-1) is a regularity condition on $G$ such that the resulting space of spanning forests is not too large. Assumption (A-2) is a commonly adopted condition on design matrix in high-dimensional linear regressions. Assumption (A-3) bounds the growth rate of the standardized true coefficients. Assumption (A-4) is a condition on the prior distributions of $\tau$ and $\sigma^2$ as well as choice of hyperparameter $c$. The proof of Theorem 1 is given in the Appendix A4.

**Theorem 1.** *(Posterior contraction) Under Assumptions (A-1) to (A-4), there exists a large enough constant $M_1 > 0$ and $\varepsilon_n \asymp \sqrt{g_n^* \log p/n}$ such that the posterior distribution satisfies $\pi_n(\|\boldsymbol{\beta} - \boldsymbol{\beta}^*\|_2 \geq M_1 \sigma^* \varepsilon_n \mid \mathbf{y}) \leq \exp(-c_1 n \varepsilon_n^2)$ with probability $1 - \exp(-c_2 n \varepsilon_n^2)$ for some constants $c_1 > 0$ and $c_2 > 0$.*

## 4 Numerical Examples

### 4.1 Simulation Studies

We conduct a simulation study to demonstrate the utility of our model. Motivated by the scalar-on-image regression problem, we consider a similar setting as Kang et al. (2018). We construct a $30 \times 30$ lattice graph which represents a structure of 2-D image. Column-standardized image predictors $\mathbf{X}_i \in \mathbb{R}^{900}, i = 1, \ldots, 100$ are generated from mean zero Gaussian process with kernel $K(x_j, x_l) = \exp(-d_{jl}/\vartheta)$ where $d_{jl}$ is the distance between pixels $j$ and $l$, $\vartheta$ is the range parameter with $\vartheta = 0$ indicating no dependence. True coefficient $\boldsymbol{\beta} \in \mathbb{R}^{900}$ is sparse (84% are zero) and has irregular cluster shapes with sharp discontinuities as shown in fig. 4(a). We let SNR $\in \{2, 4\}$ to set error variance $\sigma^2 = \mathrm{Var}(\mathbf{X}\boldsymbol{\beta})/\mathrm{SNR}$ and generate scalar responses $\boldsymbol{y} \in \mathbb{R}^{100}$ by $\boldsymbol{y} \sim \mathcal{N}(\mathbf{X}\boldsymbol{\beta}, \sigma^2 \mathbf{I})$.

We compared our model with the soft-thresholded Gaussian process (STGP, Kang et al., 2018), sparse fused lasso on graph (FL, Tibshirani et al., 2011; Wang et al., 2016), graph OSCAR (GOSCAR, Yang et al., 2012), and Bayesian graph Laplacian (BGL, Chakraborty and Lozano, 2019). We used mean squared prediction error (MSPE) with test set size 1000 to measure the predictive power and Rand index (RI) (Rand, 1971) to measure the clustering accuracy. For T-LoHo and STGP, we collected 4,000 posterior samples after 10,000 burn-in with 10 thin-in rate. For T-LoHo, we calculated the posterior median estimate of $\boldsymbol{\beta}$ and the cluster point estimate of $\Pi$ using Dahl (2006)'s method. For STGP, posterior mean estimate of $\boldsymbol{\beta}$ is used, and Rand index is calculated based on the binary classification (zero/non-zero) using posterior thresholding probabilities. For FL, the two tuning parameters are selected among the fixed candidate set of tuning parameter ratio $\gamma_{FL} \in \{0.2, 1, 5\}$ using the Bayesian information criterion. The T-LoHo and FL results presented here are when $(\tau_0, c) = (1, 0.5)$ and $\gamma_{FL} = 0.2$ respectively; the hyperparameter sensitivity analysis of T-LoHo and the detailed settings of other models are available in Appendix A5. All computations were performed on Intel E5-2690 v3 CPU with 128GB of memory.

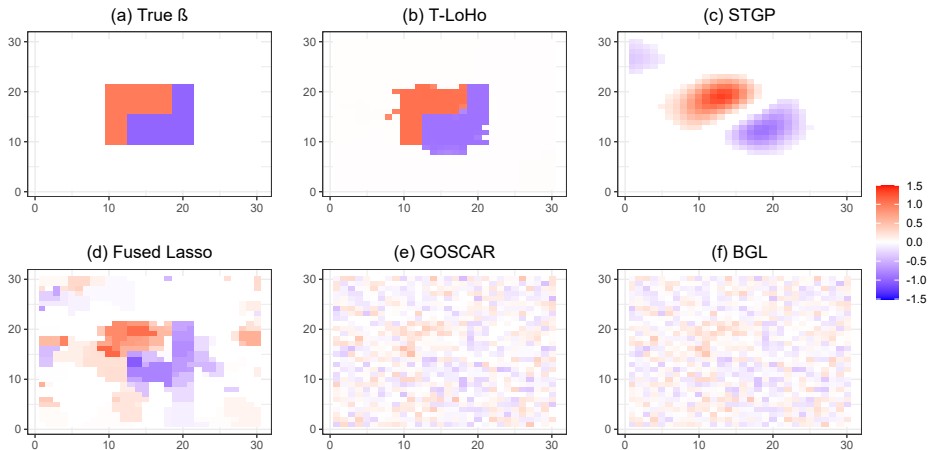

Figure 4: True and fitted results when $(\vartheta, \mathrm{SNR}) = (0, 4)$. (a) True coefficient image $\boldsymbol{\beta}$; (b) T-LoHo estimate with $(\tau_0, c) = (1, 0.5)$; (c) STGP estimate after thresholding; (d) FL estimate with tuning parameter ratio $\gamma_{FL} = 0.2$; (e) GOSCAR estimate; (f) BGL estimate.

Table 1: Performance comparison based on average MSPE and Rand index (RI) over 100 replicated simulations. Standard error is given in parentheses and time is in seconds. RI $= 1$ indicates exact recovery of the true cluster.

|       | $\vartheta$ | SNR | T-LoHo | STGP | FL | GOSCAR | BGL |
|-------|-----|-----|--------|------|-----|--------|-----|
| MSPE  | 0 | 2 | **68.5(3.0)** | 93.4(1.7) | 85.0(2.0) | 138.2(0.6) | 136.2(0.6) |
|       | 0 | 4 | **24.4(2.0)** | 86.3(1.6) | 55.8(1.4) | 133.6(0.6) | 132.3(0.6) |
|       | 3 | 2 | **251.0(11.2)** | 278.0(5.3) | 341.0(13.0) | 532.3(8.4) | 483.2(6.0) |
|       | 3 | 4 | **59.7(2.3)** | 163.9(2.2) | 115.8(3.6) | 335.0(4.8) | 213.4(2.7) |
| RI    | 0 | 2 | **0.88(0.006)** | 0.72(0.009) | 0.47(0.012) | 0.28(0.000) | 0.29(0.000) |
|       | 0 | 4 | **0.95(0.005)** | 0.72(0.010) | 0.46(0.007) | 0.28(0.000) | 0.29(0.000) |
|       | 3 | 2 | **0.87(0.004)** | 0.79(0.004) | 0.58(0.012) | 0.28(0.000) | 0.29(0.000) |
|       | 3 | 4 | **0.95(0.002)** | 0.80(0.003) | 0.57(0.010) | 0.28(0.000) | 0.29(0.000) |
| Time  | 0 | 4 | 107.9(0.4) | 339.9(1.7) | 110.4(0.6) | 0.11(0.003) | 956.2(2.3) |

From fig. 4, we can see that T-LoHo successfully captures the irregular shape of cluster boundaries and sharp discontinuities. STGP gives a much more smoothed estimate (which is expected because it does not assume homogeneity), and FL estimate contains many falsely identified non-zero clusters. The superior performance of T-LoHo over FL is partly attributed to (i) the use of horseshoe that reduces bias in FL, and (ii) the use of RSF-based prior which more efficiently searches non-zero edge differences from a spanning forest instead of a full graph $G$ used in FL. Table 1 shows that in all $(\vartheta, \mathrm{SNR})$ settings, T-LoHo indeed outperforms other models in terms of both predictive and clustering accuracy. GOSCAR and BGL give very similar results as their penalty functions have similar octagonal shapes, and both perform poorly in prediction and result in partitions with (nearly) all singletons. It is partly because GOSCAR and BGL allow the coefficients within the same cluster having similar magnitudes but with different signs. Additional simulation studies with more non-zero clusters are available in Appendix A5.

## 4.2 Anomaly Detection in Road Networks

We applied T-LoHo to the problem of detecting anomalies in road graphs. NYC Pride March is an annual event held in every June in Manhattan, New York. As the march causes traffic congestion along the route, Wang et al. (2016) considered the problem of detecting clusters on the Manhattan road network which have different taxi pickup/dropoff patterns from usual. We constructed the road network graph using GRASS GIS (GRASS Development Team, 2020) and got 3748 nodes (junctions) and 8474 edges (sections of roads). Following Wang et al. (2016), we considered the event held in

2011 (12:00–2:00 pm, June 26th) and processed the number of taxi pickup/dropoff counts data[1] to the nearest nodes with log transformation. A log baseline seasonal average was calculated from the same time block 12:00–2:00 pm on the same day of the week across the nearest 8 weeks. Then it is plausible to assume that the difference between log counts on event day and log baseline seasonal average has many zeros and clustered patterns over the graph.

We fit T-LoHo and FL and compare the results. For T-LoHo, $(\tau_0, c) = (1, 0.8)$ were used to collect 5000 posterior samples after $1.5 \times 10^5$ burn-in with 10 thin-in rate. For the FL estimate, we followed Wang et al. (2016)'s specification where the regularizaton parameter in the fused term was chosen such that the model has 200 degrees of freedom and the one in the sparsity term is set as 0.2.

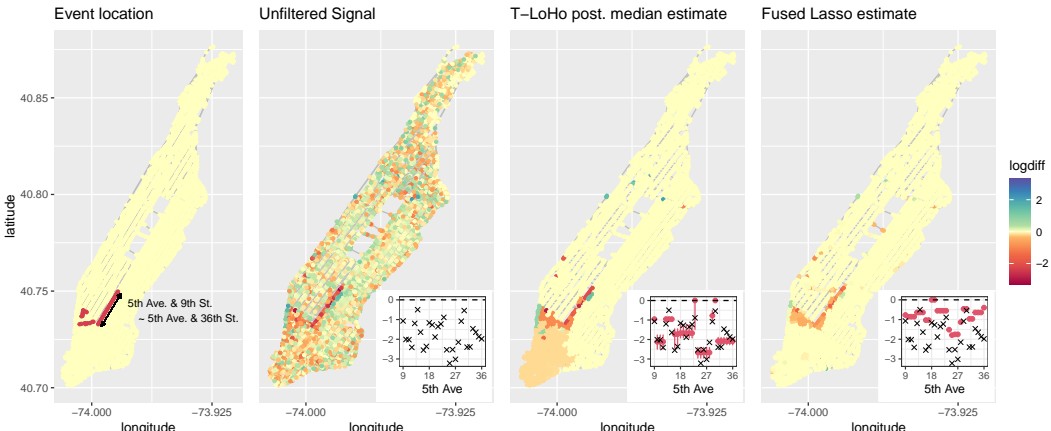

Figure 5: (Left two panels) 2011 NYC pride event route and unfiltered signal. Log-difference value below 0 indicates lower pickup/dropoff frequency than usual; (Right two panels) T-LoHo and FL estimates; (Bottom right subplots) Fitted value comparison zoomed along the parade route, 5th Ave. & 9th St. to 5th Ave. & 36th St. Here mark × indicates unfiltered signal value, red dot indicates estimated value, and red line indicates 90% credible interval for T-LoHo estimate.

Figure 5 shows the fitted results of two models. Both successfully capture the decreased taxi activity along the parade route and slightly increased taxi activity around the starting/ending point of the parade. From the subplots, we can see that FL is biased because of soft-thresholding (Rinaldo et al., 2009) while T-LoHo appears to be less biased and capable of producing reasonable uncertainty measures. A notable difference is T-LoHo can capture the decreased taxi activity around the lower Manhattan area while FL cannot due to the bias. In summary, T-LoHo gives better insight on how taxi activity changes when such event occurs.

## 5   Concluding Remarks

We propose a Tree-based Low-rank Horseshoe (T-LoHo) model to carry out Bayesian inference for graph-structured parameter which is assumed to be sparse and smooth. Accompanied with theoretical grounds and computational strategies, our simulation studies and real data example demonstrate that T-LoHo outperforms other competing methods such as fused lasso in a high-dimensional linear regression context. Extensions to other types of high-dimensional models are possible. In addition, following a similar model construction spirit as T-LoHo, we can build a general class of tree-based low-rank sparse homogeneity model extending other global-local shrinkage priors (Polson and Scott, 2010). Another scenario not addressed in this paper is when we have a weighted graph $G = (V, E, \boldsymbol{W}_0)$ as a parameter structure. In this case, incorporating *a priori* edge weight $\boldsymbol{W}_0$ to T-LoHo is a nontrivial but interesting future research question which might be useful for many possible real data applications. This work does not present any foreseeable societal consequence, but users must be fully aware of the context represented as a graph $G$ when giving interpretation on clustered parameters to avoid any misleading conclusions.

---

[1]Publicly available from NYC Taxi & Limousine Commission website, CC0.

## Acknowledgments and Disclosure of Funding

The research was partially supported by NSF DMS-1854655, NSF CCF-1934904 and NIH R01AG064010. The authors thank the referees and the area chair for valuable comments.

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
