# Appendix of "T-LoHo: A Bayesian Regularization Model for Structured Sparsity and Smoothness on Graphs"

**Changwoo J. Lee**
Department of Statistics
Texas A&M University
c.lee@stat.tamu.edu

**Zhao Tang Luo**
Department of Statistics
Texas A&M University
ztluo@stat.tamu.edu

**Huiyan Sang**
Department of Statistics
Texas A&M University
huiyan@stat.tamu.edu

The appendix consists of (1) the details of Algorithm 1 for Bayesian posterior inference in Section A1; (2) the proofs of Propositions 1 and 2 and Theorem 1 in Sections A2, A3, and A4, respectively; (3) additional simulation studies to investigate the hyper-parameter sensitivity analysis, selection criteria, and additional simulation results in Section A5.

## A1   Algorithm 1 Details

---
**Algorithm 1** One full iteration of RJMCMC posterior sampler

---
**Step 1.** Update $\Pi, K, \mathcal{F}$ using the collapsed conditional $[\Pi, K, \mathcal{F}|\Lambda, \tau, \boldsymbol{y}]$ where $\tilde{\boldsymbol{\beta}}, \sigma^2$ are integrated out. With probabilities $(p_a, p_b, p_c, p_d)$ summing up to 1, perform one of the following substeps:

   1-a. (*split*) Propose $(\Pi^\star, K^\star = K+1)$ compatible with $\mathcal{F}$, and accept with probability $\min\{1, \mathcal{A}_a \cdot \mathcal{P}_a \cdot \mathcal{L}_a\}$, where $\mathcal{A}_a$ is the prior ratio, $\mathcal{P}_a$ is the proposal ratio, $\mathcal{L}_a$ is the likelihood ratio.
   1-b. (*merge*) Propose $(\Pi^\star, K^\star = K-1)$ compatible with $\mathcal{F}$, and accept w.p. $\min\{1, \mathcal{A}_b \cdot \mathcal{P}_b \cdot \mathcal{L}_b\}$.
   1-c. (*change*) Propose $(\Pi^\star, K^\star = K)$ compatible with $\mathcal{F}$, and accept w.p. $\min\{1, \mathcal{A}_c \cdot \mathcal{P}_c \cdot \mathcal{L}_c\}$.
   1-d. (*hyper*) Update $\mathcal{F}^\star$ compatible with current $\Pi$.

**Step 2.** Jointly update $(\tau, \sigma^2, \tilde{\boldsymbol{\beta}})$ from $[\tau, \sigma^2, \tilde{\boldsymbol{\beta}} \,|\, \Lambda, \Pi, K, \mathcal{F}, \boldsymbol{y}]$, by performing:

   2-1.  Update $\tau$ from $[\tau \,|\, \Lambda, \Pi, K, \mathcal{F}, \boldsymbol{y}]$ using a Metropolis-Hastings sampler,
   2-2.  Update $\sigma^2$ from $[\sigma^2 \,|\, \tau, \Lambda, \Pi, K, \mathcal{F}, \boldsymbol{y}]$ with an inverse gamma distribution,
   2-3.  Update $\tilde{\boldsymbol{\beta}}$ from $[\tilde{\boldsymbol{\beta}} \,|\, \sigma^2, \tau, \Lambda, \Pi, K, \mathcal{F}, \boldsymbol{y}]$ with a multivariate normal distribution.

**Step 3.** Update $\Lambda$ from $[\Lambda \,|\, \tau, \sigma^2, \tilde{\boldsymbol{\beta}}, \Pi, K, \mathcal{F}, \boldsymbol{y}]$ using a slice sampler.

---

### A1.1   Step 1 Details

Step 1 is motivated by Algorithm 1 of Luo et al. (2021). It updates parameters $\Pi, K$ and $\mathcal{F}$ which determine the number of cluster and its shape. Note that step 1-d does not directly update $(\Pi, K)$ but updates the underlying MSF $\mathcal{F}$ which allows the sampler to explore different candidates of $(\Pi, K)$. Here the values of $p_a, p_b, p_c$ depend on $K$, and $p_d$ is chosen to be small enough such as 0.05 to give the sampler enough time to explore $(\Pi, K)$ compatible with current $\mathcal{F}$.

Proposal probabilities for each move $(p_a, p_b, p_c, p_d)$ depend on the current number of cluster $K$. When $n_c < K < p$, we set $(p_a, p_b, p_c, p_d) = (0.425, 0.425, 0.1, 0.05)$. For the boundary cases when $K = n_c$ and $K = p$, we set $(p_a, p_b, p_c, p_d) = (0.95, 0, 0, 0.05)$ and $(p_a, p_b, p_c, p_d) = (0, 0.95, 0, 0.05)$ respectively.

By proposition 1, there are two types of edges in $\mathcal{F} = (V, E^F)$ that determine $\Pi$: a set of cut edges (between-cluster edge) $E^C$ and a set of within-cluster edges $E^F \backslash E^C$. For step 1-a (split) with

$K^\star = K + 1$, we propose a new $\Pi^\star$ by randomly choosing one edge from the within-cluster edge set $E^F \backslash E^C$ with equal probability and switch it to a cut-edge. For step 1-b (merge) with $K^\star = K - 1$, we propose a new $\Pi^\star$ by randomly choosing one edge from the cut edge set $E^C$ with equal probability and switch it to a within-cluster edge. For step 1-c (change), $\Pi^\star$ is proposed by successively performing split and merge steps. Care is needed regarding local shrinkage parameters $\{\lambda_k\}_{k=1}^K$, since a birth/death of new cluster accompanies adding/deleting a local shrinkage parameter $\lambda^\star$. For step 1-a (split), let $\lambda^{old}$ be a local shrinkage parameter of the splitting cluster $\mathcal{C}^{old}$. By selecting a new cut-edge, we divide $\mathcal{C}^{old} = \mathcal{C}_1^{new} \cup \mathcal{C}_2^{new}$ such that $|\mathcal{C}_1^{new}| \geq |\mathcal{C}_2^{new}|$ (if two cluster sizes are the same, then based on a proposed cut-edge $(i,j)$ with $i < j$, $\mathcal{C}_1^{new}$ is the cluster that includes $i$). Then the local parameter of the bigger cluster $\mathcal{C}_1^{new}$ is inherited from the original cluster $\lambda_1^{new} = \lambda^{old}$, and the local parameter of the smaller cluster is drawn from its prior $\lambda_2^{new} = \lambda^\star \sim C^+(0,1)$. Similarly, for step 1-b (merge), we can define $\mathcal{C}^{new} = \mathcal{C}_1^{old} \cup \mathcal{C}_2^{old}$ such that $|\mathcal{C}_1^{old}| \geq |\mathcal{C}_2^{old}|$ and let $\lambda^{new} = \lambda_1^{old}$. This type of proposal makes step 1 satisfy detailed balance condition (Green, 1995).

The collapsed conditional $[\Pi, K, \mathcal{F}|\Lambda, \tau, \boldsymbol{y}]$ (where $\tilde{\boldsymbol{\beta}}$ and $\sigma^2$ are integrated out) is proportional to

$$[\Pi, K, \mathcal{F}|\Lambda, \tau, \boldsymbol{y}] \propto \iint \mathcal{N}(\boldsymbol{y}|\tilde{\mathbf{X}}\tilde{\boldsymbol{\beta}}, \sigma^2 \mathbf{I}_n) \mathcal{N}(\tilde{\boldsymbol{\beta}}|0, \sigma^2 \tau^2 \Lambda) \sigma^{-2} d\tilde{\boldsymbol{\beta}} d\sigma^2 \binom{p - n_c}{K - n_c}^{-1} (1 - c)^K$$
(A1)

$$\propto \int \mathcal{N}(\boldsymbol{y}|0, \sigma^2(\mathbf{I}_n + \tau^2 \tilde{\mathbf{X}}\Lambda\tilde{\mathbf{X}}^\top)) \times \sigma^{-2} d\sigma^2 \times \binom{p - n_c}{K - n_c}^{-1} (1 - c)^K \quad \text{(A2)}$$

$$\propto |\boldsymbol{\Sigma}|^{-1/2} (\boldsymbol{y}^\top \boldsymbol{\Sigma}^{-1} \boldsymbol{y}/2)^{-n/2} \times \binom{p - n_c}{K - n_c}^{-1} (1 - c)^K \quad \text{(A3)}$$

where $\boldsymbol{\Sigma} = \mathbf{I}_n + \tau^2 \tilde{\mathbf{X}}\Lambda\tilde{\mathbf{X}}^\top$. Note that $\tilde{\mathbf{X}} = \mathbf{X}\Phi^\top$ is a function of $\Pi$ since $\Pi$ determines $\Phi$. Line (A3) can be decomposed into a likelihood part $|\boldsymbol{\Sigma}|^{-1/2}(\boldsymbol{y}^\top \boldsymbol{\Sigma}^{-1}\boldsymbol{y}/2)^{-n/2}$ and a prior part $\binom{p - n_c}{K - n_c}^{-1}(1 - c)^K$. Then the split step proposal $(\Pi^\star, K^\star = K + 1)$ is accepted with probability $\min\{1, \mathcal{A}_a \cdot \mathcal{P}_a \cdot \mathcal{L}_a\}$ where

(prior ratio) $\quad \mathcal{A}_a = \dfrac{\binom{p - n_c}{K + 1 - n_c}^{-1}(1 - c)^{K+1} C^+(\lambda^\star|0,1)}{\binom{p - n_c}{K - n_c}^{-1}(1 - c)^K} = \dfrac{K + 1 - n_c}{p - K}(1 - c)C^+(\lambda^\star|0,1)$

(proposal ratio) $\quad \mathcal{P}_a = \dfrac{p_b}{p_a} \dfrac{\frac{1}{K+1-n_c}}{\frac{1}{(p - n_c) - (K - n_c)} C^+(\lambda^\star|0,1)} = \dfrac{p_b}{p_a} \dfrac{p - K}{(K + 1 - n_c)C^+(\lambda^\star|0,1)}$

(likelihood ratio) $\quad \mathcal{L}_a = \dfrac{|\boldsymbol{\Sigma}^\star|^{-1/2}(\boldsymbol{y}^\top \boldsymbol{\Sigma}^{\star-1}\boldsymbol{y})^{-n/2}}{|\boldsymbol{\Sigma}|^{-1/2}(\boldsymbol{y}^\top \boldsymbol{\Sigma}^{-1}\boldsymbol{y})^{-n/2}}.$

Similarly, the merge step proposal $(\Pi^\star, K^\star = K - 1)$ is accepted with probability $\min\{1, \mathcal{A}_b \cdot \mathcal{P}_b \cdot \mathcal{L}_b\}$:

(prior ratio) $\quad \mathcal{A}_b = \dfrac{\binom{p - n_c}{K - 1 - n_c}^{-1}(1 - c)^{K-1}}{\binom{p - n_c}{K - n_c}^{-1}(1 - c)^K C^+(\lambda_2^{old}|0,1)} = \dfrac{p - K + 1}{K - n_c} \dfrac{1}{(1 - c)C^+(\lambda_2^{old}|0,1)}$

(proposal ratio) $\quad \mathcal{P}_b = \dfrac{p_a}{p_b} \dfrac{\frac{1}{(p - n_c) - (K - 1 - n_c)} C^+(\lambda_2^{old}|0,1)}{\frac{1}{K - n_c}} = \dfrac{p_a}{p_b} \dfrac{(K - n_c)C^+(\lambda_2^{old}|0,1)}{p - K + 1}$

(likelihood ratio) $\quad \mathcal{L}_b = \dfrac{|\boldsymbol{\Sigma}^\star|^{-1/2}(\boldsymbol{y}^\top \boldsymbol{\Sigma}^{\star-1}\boldsymbol{y})^{-n/2}}{|\boldsymbol{\Sigma}|^{-1/2}(\boldsymbol{y}^\top \boldsymbol{\Sigma}^{-1}\boldsymbol{y})^{-n/2}},$

and the change step proposal $(\Pi^\star, K^\star = K)$ is accepted with probability $\min\{1, \mathcal{A}_c \cdot \mathcal{P}_c \cdot \mathcal{L}_c\}$:

$$\mathcal{A}_c \cdot \mathcal{P}_c \cdot \mathcal{L}_c = \dfrac{|\boldsymbol{\Sigma}^\star|^{-1/2}(\boldsymbol{y}^\top \boldsymbol{\Sigma}^{\star-1}\boldsymbol{y})^{-n/2}}{|\boldsymbol{\Sigma}|^{-1/2}(\boldsymbol{y}^\top \boldsymbol{\Sigma}^{-1}\boldsymbol{y})^{-n/2}}$$

For step 1-d (hyper), conditioning on the current partition, $\Pi$ divides the edge set $E$ of $G$ into the between-cluster edge set $E^b$ and within-cluster edge set $E^w$. We sample $\mathcal{F}$ by sampling $W_{ij} \overset{iid}{\sim}$ Unif$(0, 1/2)$ for $(i,j) \in E^w$, $W_{ij} \overset{iid}{\sim}$ Unif$(1/2, 1)$ for $(i,j) \in E^b$, and letting $\mathcal{F} = \text{MSF}(G, \boldsymbol{W})$.

To see this is a correct sampler, note that $p(\boldsymbol{W}) \propto 1$ is symmetric and $\mathcal{F} = MSF(G, \boldsymbol{W})$ depends only through the order of random edge weights $\{\boldsymbol{W}_{ij}\}_{(i,j) \in E}$; see Luo et al. (2021) for detailed discussion.

As briefly discussed in Section 3.1, we apply several computational strategies to efficiently calculate $\mathcal{L}$. Plus, if $\mathbf{X} = \mathbf{I}_n$, we can marginalize out $\Lambda$ when calculating $\mathcal{L}$ to improve MCMC mixing and avoid local parameter proposals $\lambda^\star$; see Section A1.3.

## A1.2   Step 2 and Step 3 Details

Steps 2 and 3 follow a similar procedure of Johndrow et al. (2020) as detailed below.

(2-1)  propose $\log(\tau^\star) \sim \mathcal{N}(\log(\tau), s)$, accept $\tau^\star$ w.p. $\frac{|\boldsymbol{\Sigma}^\star|^{-1/2}(\boldsymbol{y}^\top \boldsymbol{\Sigma}^{\star -1} \boldsymbol{y})^{-n/2}}{|\boldsymbol{\Sigma}|^{-1/2}(\boldsymbol{y}^\top \boldsymbol{\Sigma}^{-1} \boldsymbol{y})^{-n/2}} \frac{1+\tau^2/\tau_0^2}{1+\tau^{\star 2}/\tau_0^2} \frac{\tau^\star}{\tau}$,

(2-2)  sample $\sigma^2|\tau, \Lambda, \Pi, \boldsymbol{y} \sim \text{InvGamma}\left(\frac{n}{2}, \frac{1}{2}\boldsymbol{y}^\top \boldsymbol{\Sigma}^{-1} \boldsymbol{y}\right)$,

(2-3)  sample $\tilde{\boldsymbol{\beta}}|\sigma^2, \tau, \Lambda, \Pi, \boldsymbol{y} \sim \mathcal{N}_K\left((\tau^{-2}\Lambda^{-1} + \tilde{\mathbf{X}}^\top \tilde{\mathbf{X}})^{-1}\tilde{\mathbf{X}}^\top \boldsymbol{y}, \sigma^2(\tau^{-2}\Lambda^{-1} + \tilde{\mathbf{X}}^\top \tilde{\mathbf{X}})^{-1}\right)$,

(3)  sample $\lambda_k \overset{ind}{\sim} p(\lambda_k|\tau, \sigma^2, \tilde{\boldsymbol{\beta}}) \propto \frac{1}{1+\lambda_k^2} \times \frac{1}{\sigma\tau\lambda_k} \exp\left(-\frac{\tilde{\beta}_k^2}{2\sigma^2\tau^2\lambda_k^2}\right)$, $k = 1, \cdots, K$.

In step (2-1), the Metropolis-Hastings proposal variance $s$ is adjusted every $1000^{th}$ iteration to maintain acceptance ratio close to 0.35 (Haario et al., 2001). In step (2-3), Rue (2001)'s algorithm is used and we'll explain its benefits in the next subsection. In step (3), we use slice sampler (Neal, 2003; Polson et al., 2014) by reparametrizing $\eta_k := 1/\lambda_k^2$ and introducing auxiliary variables $\{u_k\}_{k=1}^K$. Specifically, first sample $u_k \overset{ind}{\sim} \text{Unif}(0, 1/(1 + \eta_k))$ and then sample $\eta_k \overset{ind}{\sim} \text{Exp}(2\sigma^2\tau^2/\tilde{\beta}_k^2)1(0 < \eta_k < 1/u_k - 1)$, $k = 1, \ldots, K$ where $\text{Exp}(\theta)1(0 < \eta < a)$ is a truncated exponential distribution with parameter $\theta$ (mean before truncation) and support $(0, a)$.

## A1.3   Computational Strategies

When calculating the likelihood, we utilize Sherman-Woodbury-Morrison forumla, matrix determinant lemma and Cholesky decomposition for the efficient calculation of $\boldsymbol{y}^\top \boldsymbol{\Sigma}^{-1} \boldsymbol{y}$ and $|\boldsymbol{\Sigma}|$. If we let $\mathbf{R}$ be a $K \times K$ right triangular Cholesky factor such that $\mathbf{R}^\top \mathbf{R} = \tau^{-2}\Lambda^{-1} + \tilde{\mathbf{X}}^\top \tilde{\mathbf{X}}$, then

$$\boldsymbol{y}^\top \boldsymbol{\Sigma}^{-1} \boldsymbol{y} = \boldsymbol{y}^\top \boldsymbol{y} - \boldsymbol{y}\tilde{\mathbf{X}}(\tau^{-2}\Lambda^{-1} + \tilde{\mathbf{X}}^\top \tilde{\mathbf{X}})^{-1}\tilde{\mathbf{X}}^\top \boldsymbol{y}$$
$$= \boldsymbol{y}^\top \boldsymbol{y} - \boldsymbol{y}^\top \tilde{\mathbf{X}}(\mathbf{R}^\top \mathbf{R})^{-1}\tilde{\mathbf{X}}^\top \boldsymbol{y} \tag{A4}$$

$$|\boldsymbol{\Sigma}| = |\tau^{-2}\Lambda^{-1} + \tilde{\mathbf{X}}^\top \tilde{\mathbf{X}}| \cdot |\tau^2\Lambda| = |\mathbf{R}|^2 \cdot \prod_{k=1}^{K}(\lambda_k^2\tau^2) \tag{A5}$$

It is evident to see the computational benefits of using Cholesky factor $\mathbf{R}$. First in (A4), $\boldsymbol{y}^\top \boldsymbol{y}$ is a constant and $\boldsymbol{y}^\top \tilde{\mathbf{X}}(\mathbf{R}^\top \mathbf{R})^{-1}\tilde{\mathbf{X}}^\top \boldsymbol{y} = \|\boldsymbol{v}\|_2^2$ where $\boldsymbol{v}$ is obtained by solving $\mathbf{R}^\top \boldsymbol{v} = \tilde{\mathbf{X}}^\top \boldsymbol{y}$ using only forward substitution. In (A5), $|\mathbf{R}|$ is simply a product of its diagonal elements. Next in step (2-3), using Rue (2001)'s algorithm we draw $\tilde{\boldsymbol{\beta}}$ by back substitution $\sigma^{-1}\mathbf{R}\tilde{\boldsymbol{\beta}} = (\sigma^{-1}\boldsymbol{v} + \boldsymbol{z})$ where $\boldsymbol{z} \sim \mathcal{N}_K(0, \mathbf{I}_K)$.

During the MCMC, the Cholesky factor $\mathbf{R}$ should be updated whenever a change of $\tilde{\mathbf{X}}$, $\tau$ or $\Lambda$ takes place. For instance, in step 1 we have a new $\tilde{\mathbf{X}}^\star = \mathbf{X}\Phi^{\star\top}$ which leads to $\mathbf{R}^{\star\top}\mathbf{R}^\star = \tau^{-2}\Lambda^{-1} + \tilde{\mathbf{X}}^{\star\top}\tilde{\mathbf{X}}^\star$ where a row and column are added/deleted in step 1-a/1-b respectively. A naive calculation of $\mathbf{R}$ requires $O(nK^2)$ (assuming $K < n$), but we avoid this by directly updating $\mathbf{R}$ from its previous value. Specifically, in step (1), $\tilde{\mathbf{X}}$ (and $\Lambda$) changes, and we use the rank-1 update/downdate of Cholesky factor (Golub and Van Loan, 2013, Sec. 6.5.4), (Osborne, 2010, Appendix B) which reduces the cost to $O(K^2)$. In steps (2-1) and (3), the diagonal part of $\tau^{-2}\Lambda^{-1} + \tilde{\mathbf{X}}^\top \tilde{\mathbf{X}}$ changes, and we propose a tailored updating algorithm, *Cholesky diagonal update*, and presented it in Algorithm 2. It involves about $(2/3)K^3$ FLOPs when $\mathbf{R}$ is $K \times K$. Thus, we can directly update $\mathbf{R}$ with the diagonal modification vector $\boldsymbol{v} = (\tau^{\star -2} - \tau^{-2})\text{diag}(\Lambda)^{-1}$ for step (2-1) and $\boldsymbol{v} = \tau^{-2}(\text{diag}(\Lambda^\star)^{-1} - \text{diag}(\Lambda)^{-1})$ for step 3. Combined with forward/backward substitution and Rue (2001)'s algorithm described above, these Cholesky updating strategies speed up the algorithm significantly, especially when $K$ is small.

---

**Algorithm 2** Cholesky diagonal update

---

**Input:** Right triangular Cholesky factor $chol(\boldsymbol{A}) = \boldsymbol{R} \in \mathbb{R}^{n \times n}$, diagonal part modification vector $\boldsymbol{v} \in \mathbb{R}^n$. Here $\boldsymbol{A}$ and $\boldsymbol{A} + diag(\boldsymbol{v})$ are both assumed to be positive definite.
**Output:** Right triangular Cholesky factor $\boldsymbol{U} = chol(\boldsymbol{A} + diag(\boldsymbol{v}))$.

1: Initialize $\boldsymbol{U} \in \mathbb{R}^{n \times n}$ with zero elements.
2: $\boldsymbol{w} \leftarrow \text{colSums}(\boldsymbol{R} \circ \boldsymbol{R}) + \boldsymbol{v}$ where $\circ$ indicates elementwise product.
3: **for** $i$ in 1 to $n$ **do**
4:      $U[i,i] \leftarrow \sqrt{w[i] - \sum_{k=1}^{i-1} U[k,i]^2}$
5:      $\boldsymbol{s} \leftarrow \sum_{k=1}^{i-1}(R[k,i]R[k,(i+1):n] - U[k,i]U[k,(i+1):n])$
6:      $U[i,(i+1):n] \leftarrow (R[i,i]R[i,(i+1):n] + \boldsymbol{s})/U[i,i]$
7: **end for**
8: **return** $\boldsymbol{U} = chol(\boldsymbol{A} + diag(\boldsymbol{v}))$

---

In addition, if $\mathbf{X} = \mathbf{I}_n$ we can marginalize out $\Lambda$ instead of $\sigma^2$ in step 1 which leads to improved mixing of MCMC chain and avoiding $\lambda^\star$ proposal procedure, that is, we can write a new likelihood ratio $\mathcal{L}'$ as

$$\mathcal{L}' = \frac{\int \cdots \int p(\boldsymbol{y}|\Pi^\star, K^\star, \Lambda^\star, \tau, \sigma^2) \prod_{k=1}^{K^\star} p(\lambda_k) d\lambda_k}{\int \cdots \int p(\boldsymbol{y}|\Pi, K, \Lambda, \tau, \sigma^2) \prod_{k=1}^{K} p(\lambda_k) d\lambda_k}, \tag{A6}$$

First consider $\mathcal{L}'_a$, a new likelihood ratio in the split step so that $K^\star = K + 1$ and cluster $\mathcal{C}_0$ is divided by $\mathcal{C}_0 = \mathcal{C}_1 \cup \mathcal{C}_2$. Since $\mathbf{X} = \mathbf{I}_n$, we note that likelihood $p(\boldsymbol{y}|\Pi, K, \Lambda, \tau, \sigma^2)$ can be factorized into the clusterwise likelihoods $\prod_{k=1}^{K} p(\{y\}_{i \in \mathcal{C}_k}|\Pi, K, \Lambda, \tau, \sigma^2)$ so that most of the terms in $\mathcal{L}'$ cancels out except the terms corresponding to $\mathcal{C}_0$ in the denominator and $\mathcal{C}_1, \mathcal{C}_2$ in the numerator. Thus

$$\mathcal{L}'_a = \frac{\int_0^\infty m(\{y_i\}_{i \in \mathcal{C}_1}|\lambda_1, \tau) \frac{2}{\pi(1+\lambda_1^2)} d\lambda_1 \int_0^\infty m(\{y_i\}_{i \in \mathcal{C}_2}|\lambda_2, \tau) \frac{2}{\pi(1+\lambda_2^2)} d\lambda_2}{\int_0^\infty m(\{y_i\}_{i \in \mathcal{C}_0}|\lambda_{12}, \tau) \frac{2}{\pi(1+\lambda_{12}^2)} d\lambda_{12}} \tag{A7}$$

where

$$m(\{y_i\}_{i \in C}|\lambda, \tau) = \int_{-\infty}^{\infty} \prod_{i \in C} \left[ \mathcal{N}\left(y_i|\tilde{\beta}/\sqrt{N}, \sigma^2\right) \right] \mathcal{N}\left(\tilde{\beta}|0, \sigma^2 \lambda^2 \tau^2\right) d\tilde{\beta}$$

$$= \mathcal{N}\left(\bar{y}|0, \sigma^2(1 + \lambda^2\tau^2)/N\right) \times (2\pi\sigma^2)^{-(N-1)/2} N^{-1/2} \exp(-\frac{1}{2\sigma^2} \sum_{i \in C}(y_i - \bar{y})^2)$$

for a cluster $C$ with size $N = |C|$. The three integral terms in (A7) can be calculated by numerical integration. Similarly, we can calculate the likelihood ratio of the merge step $\mathcal{L}_b$ by inverting the expression (A7), and the change step $\mathcal{L}_c$ is the product of the two. Here the product of the prior and proposal ratios remains the same as before; i.e., $\mathcal{A}' \cdot \mathcal{P}' = \mathcal{A} \cdot \mathcal{P}$ for steps (1-a), (1-b), and (1-c). Thus when $\mathbf{X} = \mathbf{I}_n$, we can update $(\Pi, K)$ using collapsed conditional $[\Pi, K|\sigma^2, \tau, \boldsymbol{y}]$ where all local parameters $(\beta_k, \lambda_k)$ are integrated out, with acceptance ratio $\alpha' = \min(1, \mathcal{A}' \cdot \mathcal{P}' \cdot \mathcal{L}')$. Computation of other parts in algorithm may also be simplified, since $\mathbf{X} = \mathbf{I}_n$ implies $\tilde{\mathbf{X}}^\top \tilde{\mathbf{X}} = \mathbf{I}_K$ and thus $\mathbf{R} = \text{diag}\left(\sqrt{\tau^{-2}\lambda_1^{-2} + 1}, \cdots, \sqrt{\tau^{-2}\lambda_K^{-2} + 1}\right)$.

## A2 Proof of Proposition 1

*Proof.* Let $G_j = (V_j, E_j)$, $j = 1, \ldots, n_c$ be connected components of $G$. Then any graph partition $\Pi = \{\mathcal{C}_1, \ldots, \mathcal{C}_K\}$ of $G$, where $K \geq n_c$, can be divided into $n_c$ disjoint subsets $\Pi = \bigcup_{j=1}^{n_c} \Pi_j$ such that each $\Pi_j$ corresponds to a graph partition of $G_j$, according to the definition of contiguous graph partitions. Note each $G_j$ is a connected graph. By applying Proposition 2 of Luo et al. (2021) for each $G_j$, there exists a spanning tree $\mathcal{T}_j = (V_j, E_j^T)$ with $|E_j^T| = |V_j| - 1$, and a set of cut-edges $E_j^C \subset E_j^T$ with $|E_j^C| = |\Pi_j| - 1$ such that $\mathcal{T}_j$ and $E_j^C$ induces $\Pi_j$. Then we can construct a spanning forest $\mathcal{F} = (V, E^F)$ where $E^F = \bigcup_{j=1}^{n_c} E_j^T$ and a set of cut-edges $E^C = \bigcup_{j=1}^{n_c} E_j^C$ with $|E^C| = \sum_{j=1}^{n_c}(|\Pi_j| - 1) = K - n_c$ which completes the proof. □

## A3 Proof of Proposition 2 and Median Variance Matching Criterion

First we present the proof of Proposition 2.

*Proof.* Here $\sigma^2, \tau_1, \tau_2$ are assumed to be fixed. If $\mu_1|\lambda_1 \sim \mathcal{N}(0, \lambda_1^2\tau_1^2\sigma^2)$ and $\mu_2|\lambda_2 \sim \mathcal{N}(0, \lambda_2^2\tau_2^2\sigma^2)$ independently, the conditional distribution of $\delta = (\mu_1 - \mu_2)/\sigma$ given $\lambda_1, \lambda_2$ is

$$\delta|\lambda_i, \lambda_j \sim \mathcal{N}(0, \lambda_1^2\tau_1^2 + \lambda_2^2\tau_2^2)$$

When $\lambda \sim C^+(0,1)$ independently, it is easy to see that density of $V = \tau^2\lambda^2$ is given by $f_V(v|\tau) = \tau/[\pi\sqrt{v}(v + \tau^2)]$, $v > 0$. Then since $\lambda_1^2\tau_1^2$ and $\lambda_2^2\tau_2^2$ are independent, the marginal distribution of $W := \lambda_1^2\tau_1^2 + \lambda_2^2\tau_2^2$ can be calculated by convolution formula; i.e.,

$$f_W(w) = \int_{-\infty}^{\infty} f_V(w - v|\tau_1)f_V(v|\tau_2)dv = \int_0^w \frac{\tau_1\tau_2}{\pi^2\sqrt{w-v}(w-v+\tau_1^2)\sqrt{v}(v+\tau_2^2)}dv$$

$$= \frac{2}{\pi^2(w+\tau_1^2+\tau_2^2)}\left[\frac{\tau_2\tan^{-1}(\frac{\tau_1\sqrt{v}}{\sqrt{w+\tau_1^2}\sqrt{w-v}})}{\sqrt{w+\tau_1^2}} + \frac{\tau_1\tan^{-1}(\frac{\sqrt{w+\tau_2^2}\sqrt{v}}{\tau_2\sqrt{w-v}})}{\sqrt{w+\tau_2^2}}\right]_{v=0}^{v\nearrow w}$$

$$= \frac{1}{\pi}\frac{\tau_1\sqrt{w+\tau_1^2} + \tau_2\sqrt{w+\tau_2^2}}{\sqrt{w+\tau_1^2}\sqrt{w+\tau_2^2}(w+\tau_1^2+\tau_2^2)}, \quad w \geq 0$$

Therefore the marginal distribution of $\delta$ can be written as

$$\pi_\Delta(\delta|\tau_1, \tau_2) = \int_0^\infty \mathcal{N}(\delta|0, w)f_W(w|\tau_1, \tau_2)dw$$

$\square$

Now we explain the median variance matching criterion under the different settings of $(n_1, n_2)$. The cumulative distribution function of $W$ is

$$F_W(w) = \frac{2}{\pi}\left[\tan^{-1}\left(\frac{\sqrt{w+\tau_2^2}}{\tau_1}\right) + \tan^{-1}\left(\frac{\sqrt{w+\tau_1^2}}{\tau_2}\right)\right] - 1$$

for $w \geq 0$ and 0 otherwise. By solving $F_W(w) = 1/2$, the median is

$$\text{median}(W) = \tau_1^2 + \tau_2^2 + 2\tau_1\tau_2\sqrt{2}$$

Since moments of $W$ do not exist, the closed form of median gives a useful measure of 'center' of $W$.

Following Section 3.2, consider the normal means model $\mathbf{X} = \mathbf{I}_n$ with two clusters $\mathcal{C}_1, \mathcal{C}_2$ with sizes $n_1, n_2$. Then T-LoHo induces a prior on the two cluster mean $\mu_1$ and $\mu_2$ as

$$\mu_1|\lambda_1, \tau, \sigma^2 \sim \mathcal{N}\left(0, \frac{1}{n_1}\lambda_1^2\tau^2\sigma^2\right), \qquad \mu_2|\lambda_2, \tau, \sigma^2 \sim \mathcal{N}\left(0, \frac{1}{n_2}\lambda_2^2\tau^2\sigma^2\right)$$

Thus when $\lambda_1, \lambda_2 \overset{iid}{\sim} C^+(0,1)$ are integrated out, the induced prior on $\delta = (\mu_1 - \mu_2)/\sigma$ is

$$\delta \sim \pi_\Delta(\delta|\tau_1 = \tau/\sqrt{n_1}, \tau_2 = \tau/\sqrt{n_2})$$

We match the median of $W$ as 1 for a fair comparison with $\mathcal{N}(0,1)$. When $n_1 = n_2 = n$ (balanced), we set $\tau = \sqrt{n}/\sqrt{2+2\sqrt{2}}$. When $n_1 : n_2 = 9 : 1$ (unbalanced), we set $\tau = \sqrt{n_1}/\sqrt{10+6\sqrt{2}}$.

## A4 Proof of Theorem 1

Let $(\boldsymbol{\beta}^*, \tilde{\boldsymbol{\beta}}^*, \sigma^*)$ denote the true $\boldsymbol{\beta}, \tilde{\boldsymbol{\beta}}$ and $\sigma$ respectively. Let $\xi^* = \{j \in V : \beta_j^* \neq 0\}$ denote the true active set of indices. Let $\breve{\Pi}$ denote an arbitrary partition of $V = \{1, \cdots, p\}$ whose corresponding partition of $\xi^*$ is determined by removing those edges with $\beta_i^* - \beta_j^* > 0$ from the subgraph of any

$\mathcal{F}$ compatible with $\breve{\Pi}$ at vertex set $\xi^*$. We define $g_n^* = \max_{\breve{\Pi}} |\breve{\Pi}(\xi^*)|$ among all possible $\breve{\Pi}$. Let $P_n$ denote all unique partitions that have at most $g_n^*(1 + c_\delta)$ clusters and their corresponding partitions of $\xi^*$ are nested in the true partition of $\xi^*$ for some constant $c_\delta > 0$.

Below, we consider the case when $p$ is much larger than $n$ and establish posterior concentration results for the T-LoHo model as $n$ goes to infinity. Our results rely on the following assumptions:

(A-1) The graph satisfies $g_n^* \prec n/\log p$, $n_c = o(g_n^*)$, and $\log |P_n| = O(g_n^* \log p)$.

(A-2) All the covariates are uniformly bounded. There exist some fixed constant $\lambda_0 > 0$, such that $\lambda_{\min}(\tilde{\mathbf{X}}^T \tilde{\mathbf{X}}) \geq n\lambda_0$ for any partition in $P_n$.

(A-3) $\max_j |\tilde{\beta}_j^*|/\sigma^* < L$, where $\log(L) = O(\log p)$.

(A-4) $-\log \tau \asymp \log p$, $\tau < p^{-(2+c_\tau)}\sqrt{g_n^* \log p / n}$, $1 - c \geq p^{-c_\alpha}$, and $\min_{\sigma^2 \in [\sigma^{*2}, \sigma^{*2}(1+c_\sigma \varepsilon_n^2)]} \pi(\sigma^2) > 0$ for some positive constants $c_\tau$, $c_\alpha$ and $c_\sigma$.

**Theorem 1.** *(Posterior contraction) Under Assumptions (A-1) to (A-4), there exists a large enough constant $M_1 > 0$ and $\varepsilon_n \asymp \sqrt{g_n^* \log p / n}$ such that the posterior distribution satisfies $\pi_n(\|\boldsymbol{\beta} - \boldsymbol{\beta}^*\|_2 \geq M_1 \sigma^* \varepsilon_n \mid \mathbf{y}) \leq \exp(-c_1 n \varepsilon_n^2)$ with probability $1 - \exp(-c_2 n \varepsilon_n^2)$ for some constants $c_1 > 0$ and $c_2 > 0$.*

### A4.1 Notations and Lemmas

We begin by introducing some asymptotic notations. Given two positive sequences $\{a_n\}$ and $\{b_n\}$, $a_n \succ b_n$ means $\lim_{n\to\infty}(a_n/b_n) = \infty$ and $a_n \asymp b_n$ means $0 < \liminf_{n\to\infty}(a_n/b_n) \leq \limsup_{n\to\infty}(a_n/b_n) < \infty$. We denote the $\ell_2$ norm by $\|\cdot\|$. We use $p$ to denote the dimension of $\boldsymbol{\beta}$.

We also introduce the following notations. Let $f_\theta$ be the likelihood function of parameter $\theta \in \Theta_n$ given $\mathbf{y}$ from a data generation model, whose prior is $\pi(\theta)$ and true parameter value is $\theta^*$. We use $f^* \equiv f_{\theta^*}$ to denote the density function of $\mathbf{y}$ when $\theta = \theta^*$, $m(\mathbf{y})$ to denote the marginal density of $\mathbf{y}$, $\mathrm{E}_\theta, \mathrm{E}^*$ to denote the expectations under $f_\theta$ and $f^*$ respectively, $\mathrm{Pr}^*$ to denote the probability measure under $p^*$, and $\pi_n$ to denote the posterior distribution given $\mathbf{y}$.

We let $\Pi(\boldsymbol{\beta})$ denote the contiguous partition of $\{1, \cdots, p\}$ with respect to $G$ according to $\boldsymbol{\beta}$. Define $\xi^1(\boldsymbol{\beta}, \sigma) := \bigcup_{\{k: |\tilde{\beta}_k/\sigma| \geq \varepsilon_n/p\}} \mathcal{C}_k$ such that it only includes the set of indices in $\{1, \cdots, p\}$ whose corresponding $|\tilde{\beta}/\sigma|$ is greater than $\varepsilon_n/p$, and $\Pi^1(\boldsymbol{\beta}, \sigma)$ be the partition of $\xi^1(\boldsymbol{\beta}, \sigma)$ under $\Pi(\boldsymbol{\beta})$. Below, we also use $\Pi(\xi)$ to denote the partition of set $\xi(\boldsymbol{\beta}, \sigma)$ when there is no risk of confusion.

We now state some lemmas that will be used in the proof of Theorem 1.

**Lemma 1.** *(Lemma 1 of Laurent and Massart (2000)) Let $\chi_d^2$ be a chi-square distribution with degree of freedom $d$. Then the following concentration inequalities hold for any $x > 0$:*

$$\mathrm{Pr}\left(\chi_d^2 > d + 2x + 2\sqrt{dx}\right) \leq \exp(-x)$$

*and*

$$\mathrm{Pr}\left(\chi_d^2 < d - 2\sqrt{dx}\right) \leq \exp(-x).$$

**Lemma 2.** *(Lemmas 10 and 11 of Banerjee (2021)) Let $\pi_{HS}(\beta|\sigma = 1, \tau)$ denote the horseshoe prior density on $\beta$ assuming a fixed $\tau > 0$ and $\sigma^2 = 1$. Under Assumptions (A-3) and (A-4), for some $c_\tau > 0$, we have*

$$\int_{-\varepsilon_n/p}^{\varepsilon_n/p} \pi_{HS}(\beta; 1, \tau)d\beta \geq 1 - p^{-(1+c_\tau)}, \quad -\log\left(\inf_{\beta \in [-L, L]} \pi_{HS}(\beta; 1, \tau)\right) = O(\log p) \quad \text{(A8)}$$

**Lemma 3.** *(Lemma A.3 of Song and Cheng (2020)) Let $B_n$ and $C_n$ be two subsets of the parameter space $\Theta_n$, and $\phi_n$ be a test function satisfying $\phi_n(D_n) \in [0, 1]$ for any data $D_n$. If $\pi(B_n) \leq b_n, \mathrm{E}^*\{\phi_n(D_n)\} \leq b_n', \sup_{\theta \in C_n} \mathrm{E}\{1 - \phi_n(D_n)\} \leq c_n$, and*

$$\mathrm{Pr}^*\left(\frac{m(D_n)}{f^*(D_n)} \geq a_n\right) \geq 1 - a_n'.$$

*Then*

$$\mathrm{E}^*\{\pi_n(C_n \cup B_n \mid D_n)\} \leq \frac{b_n + c_n}{a_n} + a_n' + b_n'.$$

### A4.2 Proof of Theorem 1

*Proof.* The proof of Theorem 1 proceeds by verifying the conditions in Lemma 3 in three parts, adapting the proof in Song and Liang (2017) originally used for a Bayesian variable selection problem. The first part involves the construction of $B_n$ and $C_n$, the second part shows the existence of testing functions $\phi_n$, and the last part proves the evidence lower bound.

**Part 1** (Sieve construction): We define

$$C_n = \left\{ (\boldsymbol{\beta}, \sigma) : \|\boldsymbol{\beta} - \boldsymbol{\beta}^*\| \leq M_1 \sigma^* \varepsilon_n \text{ and } \frac{1 - \varepsilon_n}{1 + \varepsilon_n} < \sigma^2 / \sigma^{*2} < \frac{1 + \varepsilon_n}{1 - \varepsilon_n} \right\}^c \setminus B_n,$$

and

$$B_n = \left\{ (\boldsymbol{\beta}, \sigma) : \Pi^1(\boldsymbol{\beta}, \sigma) \text{ has at least } c_\delta g_n^* \text{ clusters } \right\}.$$

for some constant $c_\delta > 0$.

Recall $K \equiv |\Pi(\boldsymbol{\beta}, \sigma)|$ follows a left truncated geometric distribution. Conditional on $K$, the number of clusters whose corresponding $|\tilde{\beta}/\sigma|$ exceeds $\varepsilon_n/p$ follows a $\text{Binomial}(K, v_n)$ distribution, where $v_n = \int_{|x| \geq \varepsilon_n/p} \pi_{hs}(x; 1, \tau) dx$ and $\pi_{hs}(x)$ is the horseshoe prior density function for $\tilde{\beta}/\sigma$. From Lemma 2, it is known that $v_n \leq O(p_n^{-(1+c_\tau)})$. Following a similar proof as in Song and Liang (2017), we can show that for any $K > c_\delta g_n^*$, $\Pr(\text{Binomial}(K, v_n) \geq c_\delta g_n^*) \leq \exp(-c_1' n \epsilon_n^2)$ for some constant $c_1'$ using a sharp tail bound for binomial distributions in Theorem 1 of Zubkov and Serov (2013). Conditional on any given spanning forest $\mathcal{F}$, $\pi(B_n \mid \mathcal{F}) = \Pr(\text{Binomial}(K, v_n) \geq c_\delta g_n^* \mid K \geq c_\delta g_n^*) \Pr(K \geq c_\delta g_n^*) \leq \exp(-c_1' n \varepsilon_n^2)$. Note that $c_1'$ does not depend on $\mathcal{F}$. Therefore there exists some constant $c_1$ such that

$$\pi(B_n) = \sum_{\mathcal{F}} \pi(B_n \mid \mathcal{F}) \pi(\mathcal{F}) \leq \exp\{-c_1 n \epsilon_n^2\}. \tag{A9}$$

**Part 2** (Existence of testing function), Given an arbitrary partition $\Pi(\xi)$ of an arbitrary set $\xi$ with $K_\xi$ clusters, let $\tilde{\mathbf{X}}_\xi$ be the corresponding transformed design matrix, and $\mathbf{H}_\xi$ be the hat matrix of $\tilde{\mathbf{X}}_\xi$. Define $\hat{\boldsymbol{\beta}}_\xi = (\tilde{\mathbf{X}}_\xi^T \tilde{\mathbf{X}}_\xi)^{-1} \tilde{\mathbf{X}}_\xi^T \mathbf{y}$, $\hat{\sigma}_\xi^2 = \mathbf{y}^T (\mathbf{I} - \mathbf{H}_\xi) \mathbf{y} / (n - K_\xi)$, and $\tilde{\boldsymbol{\beta}}_\xi^* = \Phi(\Pi(\xi)) \beta^*(\xi)$.

We define a test function

$$\phi(\mathbf{y}) = \max_{\{\xi \supset \xi^*\}} \mathbf{1}\{\|\hat{\boldsymbol{\beta}}_\xi - \tilde{\boldsymbol{\beta}}_\xi^*\| \geq \sigma^* \varepsilon_n \text{ and } |\hat{\sigma}_\xi^2 / \sigma^{*2} - 1| \geq \varepsilon_n, \text{ for some } \xi \text{ and } \Pi(\xi)$$

$$\text{such that } \Pi(\xi^*) \text{ is nested in } \Pi^*(\xi^*) \text{ and } K_\xi \leq (1 + c_\delta) g_n^* \}$$

for some fixed $c_\delta > 0$, where $\Pi^*(\xi^*)$ is the true partition of $\xi^*$.

From standard linear regression results, under the true parameter $(\boldsymbol{\beta}^*, \sigma^*)$ and the restricted eigenvalue assumption in (A-2), we have $\Pr(|\hat{\sigma}_\xi^2 / \sigma^{*2} - 1| \geq \varepsilon_n) = \Pr\left(|\chi_{n-K_\xi}^2 / (n - K_\xi) - 1| \geq \varepsilon_n\right)$ and $\Pr(\|\hat{\boldsymbol{\beta}}_\xi - \tilde{\boldsymbol{\beta}}_\xi^*\| \geq \sigma^* \varepsilon_n) \leq \Pr\left(\chi_{K_\xi}^2 \geq n \lambda_0 \varepsilon_n^2\right)$. Using Lemma 1, we have $\Pr(\|\hat{\boldsymbol{\beta}}_\xi - \tilde{\boldsymbol{\beta}}_\xi^*\| \geq \sigma^* \varepsilon_n \text{ and } |\hat{\sigma}_\xi^2 / \sigma^{*2} - 1| \geq \varepsilon_n) \leq \exp(-c_{21}' n \varepsilon_n^2)$. From the union bound and the last part of Assumption (A-1), we can now bound the type-I error of the test function as follows

$$\mathrm{E}^* \{\phi(\mathbf{y})\} \leq P_n \cdot \exp(-c_{21}' n \varepsilon_n^2) \leq \exp(-c_{21} n \varepsilon_n^2), \tag{A10}$$

for some constant $c_{21} > 0$ and large $n \varepsilon_n^2 / (g_n^* \log p)$.

Next we bound the type II error part $\sup_{(\beta, \sigma) \in C_n} \mathrm{E}\{1 - \phi(\mathbf{y})\}$. We define two subsets $C_n^{(1)}$ and $C_n^{(2)}$ such that $C_n \subset C_n^{(1)} \cup C_n^{(2)}$ and analyze them separately, where

$$C_n^{(1)} = \left\{ (\boldsymbol{\beta}, \sigma) : \|\boldsymbol{\beta} - \boldsymbol{\beta}^*\| > M_1 \sigma^* \varepsilon_n, \frac{\sigma^2}{\sigma^{*2}} < \frac{1 + \varepsilon_n}{1 - \varepsilon_n} \right\} \cap B_n^c$$

and

$$C_n^{(2)} = \left\{ \sigma : \frac{\sigma^2}{\sigma^{*2}} \leq \frac{1 - \varepsilon_n}{1 + \varepsilon_n} \text{ or } \frac{\sigma^2}{\sigma^{*2}} \geq \frac{1 + \varepsilon_n}{1 - \varepsilon_n} \right\} \cap B_n^c.$$

For any $\boldsymbol{\beta} \in C_n$, let $\mathcal{F}(\boldsymbol{\beta})$ be any spanning forest that can induce $\Pi(\boldsymbol{\beta})$. Consider a set $\breve{\xi}$ formed by keeping those column indices whose $|\tilde{\beta}/\sigma| > \varepsilon_n/p$ or $\beta^* > 0$. Thus $\breve{\xi} \supset \xi^*$. We then form a partition $\Pi(\breve{\xi})$ of $\breve{\xi}$ by removing those edges with $\beta_i^* - \beta_j^* > 0$ or $\beta_i - \beta_j > 0$ from the subgraph of $\mathcal{F}(\boldsymbol{\beta}, \sigma)$ at vertex set $\breve{\xi}$. By construction and the definition of $B_n$, $\Pi(\xi^*)$ is nested in $\Pi^*(\xi^*)$ and $K_{\breve{\xi}} \leq (1 + c_\delta)g_n^*$.

For any $(\boldsymbol{\beta}, \sigma) \in C_n^{(1)}$, we can show that

$$\Pr\left\{\|\hat{\boldsymbol{\beta}}_{\breve{\xi}} - \tilde{\boldsymbol{\beta}}_{\breve{\xi}}^*\| \leq \sigma^* \varepsilon_n \mid \boldsymbol{\beta}, \sigma^2\right\}$$

$$= \Pr\left\{\|(\tilde{\mathbf{X}}_{\breve{\xi}}^T \tilde{\mathbf{X}}_{\breve{\xi}})^{-1}\tilde{\mathbf{X}}_{\breve{\xi}}^T \sigma z + \tilde{\boldsymbol{\beta}}_{\breve{\xi}} + (\tilde{\mathbf{X}}_{\breve{\xi}}^T \tilde{\mathbf{X}}_{\breve{\xi}})^{-1}\tilde{\mathbf{X}}_{\breve{\xi}}^T \tilde{\mathbf{X}}_{\breve{\xi}^c}\tilde{\boldsymbol{\beta}}_{\breve{\xi}^c} - \tilde{\boldsymbol{\beta}}_{\breve{\xi}}^*\| \leq \sigma^* \varepsilon_n\right\}$$

$$\leq \Pr\left\{\|(\tilde{\mathbf{X}}_{\breve{\xi}}^T \tilde{\mathbf{X}}_{\breve{\xi}})^{-1}\tilde{\mathbf{X}}_{\breve{\xi}}^T z\| \geq \left[\|\tilde{\boldsymbol{\beta}}_{\breve{\xi}} - \tilde{\boldsymbol{\beta}}_{\breve{\xi}}^*\| - \|(\tilde{\mathbf{X}}_{\breve{\xi}}^T \tilde{\mathbf{X}}_{\breve{\xi}})^{-1}\tilde{\mathbf{X}}_{\breve{\xi}}^T \tilde{\mathbf{X}}_{\breve{\xi}^c}\tilde{\boldsymbol{\beta}}_{\breve{\xi}^c}\| - \sigma^* \varepsilon_n\right]/\sigma\right\}$$

$$\leq \Pr\left\{\|(\tilde{\mathbf{X}}_{\breve{\xi}}^T \tilde{\mathbf{X}}_{\breve{\xi}})^{-1}\tilde{\mathbf{X}}_{\breve{\xi}}^T z\| \geq M_2 \varepsilon_n\right\} \tag{A11}$$

$$\leq \exp(-c_{22}' n \varepsilon_n^2) \tag{A12}$$

where $z := \epsilon/\sigma$ following $N(0,1)$. The last inequality in (A12) is from Lemma 1. The second last inequality in (A11) holds because $\|\tilde{\boldsymbol{\beta}}_{\breve{\xi}} - \tilde{\boldsymbol{\beta}}_{\breve{\xi}}^*\| \geq \|\boldsymbol{\beta}_{\breve{\xi}} - \boldsymbol{\beta}_{\breve{\xi}}^*\| - \|\tilde{\boldsymbol{\beta}}_{\breve{\xi}^c}\| \geq M_1 \sigma^* \varepsilon_n - p_n(\sigma \varepsilon_n/p) \geq M_1 \sigma^* \varepsilon_n - \sigma^* \varepsilon_n \sqrt{(1 + \varepsilon_n)/(1 - \varepsilon_n)}$, and $\|(\tilde{\mathbf{X}}_{\breve{\xi}}^T \tilde{\mathbf{X}}_{\breve{\xi}})^{-1}\tilde{\mathbf{X}}_{\breve{\xi}}^T \tilde{\mathbf{X}}_{\breve{\xi}^c}\tilde{\boldsymbol{\beta}}_{\breve{\xi}^c}\| \leq \sqrt{1/(n\lambda_0)}\sqrt{np}\sqrt{p}\sigma \varepsilon_n/p \leq c_{23}'\sigma \varepsilon_n$ by Assumption (A-2).

For any $(\boldsymbol{\beta}, \sigma) \in C_n^{(2)}$, from linear regression results, we have $\left\|\mathbf{y} - \tilde{\mathbf{X}}_{\breve{\xi}}\hat{\boldsymbol{\beta}}_{\breve{\xi}}\right\|^2 \sim \sigma^2 \chi_{n-K_{\breve{\xi}}}^2(\kappa)$, where the noncentral parameter $\kappa = \|\tilde{\mathbf{X}}_{\breve{\xi}^c}\boldsymbol{\beta}_{\breve{\xi}^c}/\sigma\|_2^2 \leq c_\kappa n \varepsilon_n^2$. Therefore,

$$\Pr_{(\beta,\sigma)}\left(\left|\hat{\sigma}_{\hat{\pi}}^2(\mathbf{y}) - \sigma^{*2}\right| < \sigma^{*2}\varepsilon_n\right)$$

$$= \Pr_{(\beta,\sigma)}\left(\left|\frac{\left\|\mathbf{y} - \tilde{\mathbf{X}}_{\breve{\xi}}\hat{\boldsymbol{\beta}}_{\breve{\xi}}\right\|^2}{\sigma^{*2}(n - K_{\breve{\xi}})} - 1\right| < \varepsilon_n\right)$$

$$\leq \Pr_{(\beta,\sigma)}\left(\left|\frac{\left\|\mathbf{y} - \tilde{\mathbf{X}}_{\breve{\xi}}\hat{\boldsymbol{\beta}}_{\breve{\xi}}\right\|^2}{\sigma^2} - (n - K_{\breve{\xi}})\right| > (n - K_{\breve{\xi}})\varepsilon_n\right)$$

$$\leq \Pr\left(\left|\chi_{n-K_{\breve{\xi}}}^2(\kappa) - (n - K_{\breve{\xi}})\right| > (n - K_{\breve{\xi}})\varepsilon_n\right)$$

$$\leq \exp\left(-c_{24}' n \varepsilon_n^2\right), \tag{A13}$$

for some constant $c_{24}' > 0$ and large $n$.

Therefore,

$$\sup_{(\beta,\sigma)\in C_n} \mathrm{E}_{(\beta,\sigma)}\{1 - \phi(y)\} \leq \max\{\sup_{(\beta,\sigma)\in C_n^{(1)}} \mathrm{E}_{(\beta,\sigma)}\{1 - \phi(y)\}, \sup_{(\beta,\sigma)\in C_n^{(2)}} \mathrm{E}_{(\beta,\sigma)}\{1 - \phi(y)\}\}$$

$$\leq \max\left\{\exp\left(-c_{22}' n \varepsilon_n^2\right), \exp\left(-c_{24}' n \varepsilon_n^2\right)\right\}$$

$$\leq \exp\left(-c_{22} n \varepsilon_n^2\right), \tag{A14}$$

**Part 3** (Evidence lower bound): Recall $m(\mathbf{y})$ is the marginal likelihood and $f^*(\mathbf{y})$ is the true likelihood. We let $\boldsymbol{\epsilon}^* = \mathbf{y} - \mathbf{X}\boldsymbol{\beta}^*$ be the vector of error terms. Under our model,

$$\frac{m(\mathbf{y})}{f^*(\mathbf{y})} = \int \exp(R_n)\pi(\boldsymbol{\beta}, \sigma^2)d\boldsymbol{\beta}d\sigma^2,$$

where $R_n$ is the log likelihood ratio $-\frac{1}{2\sigma^2}\|\mathbf{X}\boldsymbol{\beta}^* + \boldsymbol{\epsilon}^* - \mathbf{X}\boldsymbol{\beta}\|^2 + \frac{\|\boldsymbol{\epsilon}^*\|^2}{2\sigma^{*2}} - n\log\frac{\sigma}{\sigma^*}$.

In Part 3, we seek to prove

$$\Pr^*\{\frac{m(\mathbf{y})}{f^*(\mathbf{y})} \geq \exp(-c_{31}n\varepsilon_n^2)\} \geq 1 - \exp(-c_{32}n\varepsilon_n^2), \tag{A15}$$

for some constants $c_{31}$ and $c_{32}$.

We will first show that (A15) holds under the event $E_1 = \{\|\boldsymbol{\epsilon}^*/\sigma^*\|^2 \le (1+c'_{31})n$ and $\|\boldsymbol{\epsilon}^{*T}\mathbf{X}\|_\infty \le c'_{31}n\varepsilon_n\}$ and then show that $E_1$ holds with a large probability.

To prove (A15), rewrite

$$R_n = \underbrace{-\frac{\|\mathbf{X}\boldsymbol{\beta}^* - \mathbf{X}\boldsymbol{\beta}\|^2}{2\sigma^2}}_{I} \underbrace{-\frac{(\mathbf{X}\boldsymbol{\beta}^* - \mathbf{X}\boldsymbol{\beta})^\top \boldsymbol{\epsilon}^*}{\sigma^2}}_{II} + \underbrace{\|\boldsymbol{\epsilon}^*\|^2\left(\frac{1}{2\sigma^{*2}} - \frac{1}{2\sigma^2}\right)}_{>0} \underbrace{-\frac{n}{2}\log\frac{\sigma^2}{\sigma^{*2}}}_{>-\delta'_{31}n\varepsilon_n^2/2} \quad \text{(A16)}$$

Under the event of $E_1$, when $H_2 := \left\{\sigma^2 \in [\sigma^{*2},\ \sigma^{*2}(1+\delta'_{31}\varepsilon_n^2)] \text{ and } \|(\boldsymbol{\beta}^* - \boldsymbol{\beta})/\sigma\|_1 < 2\delta'_{32}\varepsilon_n\right\}$ holds for some constants $\delta'_{31}$ and $\delta'_{32}$, we can show that $I \ge O(-n\varepsilon_n^2)$ and $II \ge O(-n\varepsilon_n^2)$ using inequality between $\ell_1$ and $\ell_2$ norms and Hölder inequality. Thus, $H_1 := \{R_n \ge -c'_{32}n\varepsilon_n^2\}$ is a super-set of $H_2$ for some constant $c'_{32}$.

Let $\pi(\sigma^2)$ denote the prior density function for $\sigma^2$. By Assumption (A-4), for some constant $c_\sigma > 0$,

$$\Pr\left\{\sigma^2 \in [\sigma^{*2},\ \sigma^{*2}(1+\delta'_{31}\varepsilon_n^2)]\right\}$$
$$\ge \delta'_{31}\sigma^{*2}\varepsilon_n^2 \cdot \min_{\sigma^2\in[\sigma^{*2},\ \sigma^{*2}(1+\delta'_{31}\varepsilon_n^2)]}\pi(\sigma^2)$$
$$\ge \exp(-c'_\sigma n\varepsilon_n^2). \quad \text{(A17)}$$

Let $\tilde{\Pi}$ denote an arbitrary partition of $\{1,\ldots,p\}$ whose corresponding partition of $\xi^*$ is determined by removing those edges with $\beta_i^* - \beta_j^* > 0$ from the subgraph of any $\mathcal{F}$ compatible with $\tilde{\Pi}$ at vertex set $\xi^*$. Also let $\tilde{K}$, $\tilde{\boldsymbol{\beta}}$ and $\tilde{\beta}^*$ denote its number of clusters, the transformed coefficients and true coefficients, respectively. We now analyze the prior of $\tilde{\boldsymbol{\beta}}$. Without loss of generality, assume the first $\tilde{k}^1$ clusters of $\tilde{\Pi}$ have nonzero $\boldsymbol{\beta}^*$. We write $\|(\boldsymbol{\beta}^* - \boldsymbol{\beta})/\sigma\|_1 = \sum_{j=1}^{\tilde{k}^1}\sqrt{|\mathcal{C}_j|}\cdot|\tilde{\beta}_j - \tilde{\beta}_j^*|/\sigma + \sum_{j=\tilde{k}^1+1}^{\tilde{K}}\sqrt{|\mathcal{C}_j|}\cdot|\tilde{\beta}_j|/\sigma$. Therefore,

$$\{\|(\boldsymbol{\beta}^* - \boldsymbol{\beta})/\sigma\|_1 < 2\delta'_{32}\varepsilon_n\} \supset \left\{|\tilde{\beta}_j/\sigma| \le \delta'_{32}\varepsilon_n/\sqrt{\tilde{K}p}, \text{ for all } j = \tilde{k}^1+1,\cdots,\tilde{K}\right\} \cap$$

$$\left\{\tilde{\beta}_j/\sigma \in [\tilde{\beta}_j^*/\sigma - \delta'_{32}\varepsilon_n/\sqrt{\tilde{k}^1|\xi^*|}, \tilde{\beta}_j^*/\sigma + \delta'_{32}\varepsilon_n/\sqrt{\tilde{k}^1|\xi^*|}] \text{ for all } j = 1,\cdots,\tilde{k}^1\right\} \text{ from the in-}$$

equality between $l_1$ and $l_2$ norms.

Note that $\tilde{k}^1 \le g_n^*$ and $\tilde{K} \le p$. Conditional on $\sigma^2 \in [\sigma^{*2}, \sigma^{*2}(1+\delta'_{31}\varepsilon_n^2)]$ and $\tilde{\Pi}$, we have

$$\Pr\left\{|\tilde{\beta}_j/\sigma| \le \delta'_{32}\varepsilon_n/\sqrt{\tilde{K}p}, \text{ for all } j = \tilde{k}^1+1,\cdots,\tilde{K}\right\} \ge (1 - p^{-1-c_{hs}})^{\sqrt{\tilde{K}p}} \to 1 \quad \text{(A18)}$$

and

$$\Pr\left\{\tilde{\beta}_j/\sigma \in [\tilde{\beta}_j^*/\sigma - \delta'_{32}\varepsilon_n/\sqrt{\tilde{k}^1|\xi^*|}, \tilde{\beta}_j^*/\sigma + \delta'_{32}\varepsilon_n/\sqrt{\tilde{k}^1|\xi^*|}] \text{ for all } j = 1,\cdots,\tilde{k}^1\right\}$$

$$\ge \left[2\delta'_{32}\varepsilon_n\left(\inf_{\tilde{\beta}/\sigma\in[-L,L]}\pi_{HS}(\beta;1,\tau)\right)/\sqrt{\tilde{k}^1|\xi^*|}\right]^{\tilde{k}^1} \ge \exp(-c'_\beta n\varepsilon_n^2) \quad \text{(A19)}$$

from Lemma 2 and Assumption (A-1). Combining (A17), (A18) and (A19), we can show that $\Pr(H_2|\tilde{\Pi}) \ge \exp\left(-c'_{33}n\varepsilon_n^2\right)$.

Also notice $\Pr(\tilde{\Pi} \mid \mathcal{F}) = \sum_{\tilde{K}\ge\tilde{k}^1+n_c-n_c^*}\Pr(k = \tilde{K})\binom{p-n_c-\tilde{k}^1+n_c^*}{\tilde{K}-n_c-\tilde{k}^1+n_c^*}\binom{p-n_c}{\tilde{K}-n_c}^{-1}$, where $n_c^*$ is the number of components in the original graph on $\xi^*$. By Assumption (A-1),

$$\log\Pr(k = \tilde{k}^1 + n_c - n_c^*) \ge \log\frac{(1-c)^{2g_n^*}}{\sum_{k=1}^{p_n}(1-c)^k}$$
$$= (2g_n^* - 1)\log(1-c) + \log c - \log\{1 - (1-c)^p\}$$
$$\ge -2c_\alpha g_n^*\log p. \quad \text{(A20)}$$

In addition,

$$-\log\binom{p - n_c}{\tilde{k}^1 - n_c^*} \geq -g_n^* \log p. \tag{A21}$$

Therefore, $\Pr(H_2|\mathcal{F}) \geq \Pr(H_2 \mid \tilde{\Pi}) \Pr(\tilde{\Pi} \mid \mathcal{F}) \geq \exp\left(-c_{34}' n\varepsilon_n^2\right)$. Since $c_{34}'$ does not depend on the choice of $\mathcal{F}$, we have $\Pr(H_2) \geq \exp\left(-c_{34}' n\varepsilon_n^2\right)$.

Recall under $E_1$, $H_2$ holds and thus $R_n \geq c_{32}' n\varepsilon_n^2$ since $H_1 \supset H_2$. It follows that,

$$\begin{aligned}
\frac{m(\mathbf{y})}{f^*(\mathbf{y})} &\geq \Pr(H_2) \exp(R_n) \\
&\geq \exp(-c_{32}' n\varepsilon_n^2) \Pr(H_2) \geq \exp(-c_{31} n\varepsilon_n^2).
\end{aligned} \tag{A22}$$

Finally, from Lemma 1 and the tail bound for maximum of sub-Gaussian random variables, we can prove that $\Pr(E_1) \geq 1 - \exp\{-c_{32} n\varepsilon_n^2\}$. Combining with (A22), we proved (A15).

**Combining three parts**: Finally, we use the results of (A9) from Part 1, (A10) and (A14) from Part 2, (A15) from Part 3 and apply Lemma 3 to obtain

$$\begin{aligned}
&\mathrm{E}^* \left\{ \pi_n \left( \|\boldsymbol{\beta} - \boldsymbol{\beta}^*\|_2 \geq M_1 \sigma^* \varepsilon_n \mid \mathbf{y} \right) \right\} \\
&\leq \mathrm{E}^* \left\{ \pi_n \left( C_n \cup B_n \mid \mathbf{y} \right) \right\} \leq \exp(-c_e n\varepsilon_n^2)
\end{aligned}$$

for some constant $c_e > 0$. The result in Theorem 1 then follows from Markov inequality and Borel-Cantelli lemma, which completes the proof of Theorem 1.

$\square$

## A5 Hyperparameter Selection and Computational Efficiency Analysis

### A5.1 Hyperparameter Sensitivity Analysis

First we present additional simulation study results in Section 4.1 with different sets of hyperparameters. For T-LoHo, 6 choices of $(\tau_0, c) \in \{0.1, 1\} \times \{0.1, 0.5, 0.9\}$ are considered. Note that we used posterior median estimate for $\hat{\boldsymbol{\beta}}^{TLoHo}$, because T-LoHo gives multimodal posterior distribution near the cluster boundary where using posterior mean can lead to misleading summary. For soft-thresholded GP (STGP) (Kang et al., 2018), we used $15 \times 15$ equally spaced grid of knots with bandwidth set to the minimum distance between knots and same prior specification of section 4.1 therein. We used posterior mean to get $\hat{\boldsymbol{\beta}}^{STGP}$, where $\hat{\beta}_j^{STGP} = 0$ if posterior probability of a nonzero $\hat{\beta}_j^{STGP}$ is less than $1/2$. We ran total 50,000 MCMC iterations with 10,000 burn-in and 10 thin-in rate to collect 4,000 posterior samples for both T-LoHo and STGP. For sparse Fused lasso (FL) (Tibshirani et al., 2011), we used path algorithm (Arnold and Tibshirani, 2020) to get FL estimate

$$\hat{\boldsymbol{\beta}}_{FL} = \text{argmin}_{\boldsymbol{\beta}} \left\{ 0.5\|\boldsymbol{y} - \mathbf{X}\boldsymbol{\beta}\|_2^2 + \lambda_{FL} \sum_{(j,l) \in E} |\beta_j - \beta_l| + \gamma_{FL} \cdot \lambda_{FL} \sum_{j=1}^{p} |\beta_j| \right\} \quad \text{(A23)}$$

where we considered 3 candidates of $\gamma_{FL} \in \{0.2, 1, 5\}$ and selected $\lambda_{FL}$ among 2000 number of steps using Bayesian information criteria. For graph octagonal shrinkage and clustering algorithm for regression (GOSCAR) (Yang et al., 2012), $\lambda_1$ ($\ell_1$ regularizaton parameter) and $\lambda_2$ (pairwise $\ell_\infty$ regularization parameter) are both set as $0.8 \max\{|\beta_i|\}/|E|$ from the true value of $\max\{|\beta_i|\}$. For Bayesian graph Laplacian (Liu et al., 2014), we used EM algorithm of Chakraborty and Lozano (2019) with same prior specification of Section 2.6 therein.

We report mean squared prediction error (MSPE) $\frac{1}{n_{test}}\|\mathbf{X}_{test}\hat{\boldsymbol{\beta}} - \mathbf{X}_{test}\boldsymbol{\beta}\|^2$ based on a test set with size $n_{test} = 1000$ and Rand index (RI) (Rand, 1971) which measures the clustering accuracy. Table A1 shows comparison results with different levels of dependencies and SNR. Based on 100 replicated simulations, average MSPE and RI values are reported under 6 different settings of $(\tau_0, c)$ for T-LoHo, one STGP, and 3 different settings of $\gamma_{FL}$ for FL. In Section 4.1, we report T-LoHo result with $(\tau_0, c) = (1, 0.5)$ and FL result with $\gamma_{FL} = 0.2$.

Table A1: Performance comparison based on average MSPE and RI over 100 replicated simulations. Standard error is given in parentheses. RI= 1 indicates exact recovery of the true cluster.

| $\vartheta$, SNR | | T-LoHo with different $(\tau_0, c)$ settings | | | | | |
| --- | --- | --- | --- | --- | --- | --- | --- |
| | | (0.1, 0.1) | (0.1, 0.5) | (0.1, 0.9) | (1, 0.1) | (1, 0.5) | (1, 0.9) |
| MSPE | 0, 2 | 57.1(2.6) | 75.1(3.9) | 101.2(4.4) | **52.1(2.2)** | 68.5(3.0) | 96.3(43.3) |
| | 0, 4 | 27.6(1.4) | 30.8(1.6) | 46.5(4.1) | 27.1(1.1) | **24.4(2.0)** | 27.3(2.0) |
| | 3, 2 | **220.1(10.8)** | 241.3(11.2) | 291.5(13.1) | 235.2(10.8) | 250.9(11.2) | 296.3(14.0) |
| | 3, 4 | **55.5(2.2)** | 60.4(2.6) | 66.6(3.3) | 58.4(2.7) | 59.7(2.3) | 70.3(3.1) |
| RI | 0, 2 | 0.43(0.007) | 0.85(0.009) | 0.81(0.009) | 0.58(0.018) | **0.88(0.005)** | 0.82(0.009) |
| | 0, 4 | 0.49(0.010) | 0.95(0.003) | 0.91(0.008) | 0.70(0.022) | **0.95(0.002)** | 0.95(0.005) |
| | 3, 2 | 0.44(0.009) | **0.88(0.004)** | 0.86(0.005) | 0.71(0.020) | 0.87(0.004) | 0.86(0.005) |
| | 3, 4 | 0.52(0.013) | 0.94(0.002) | 0.94(0.002) | 0.89(0.013) | **0.95(0.002)** | 0.94(0.002) |

| $\vartheta$, SNR | | STGP | FL with different $\gamma_{FL}$ settings | | | GOSCAR | BGL |
| --- | --- | --- | --- | --- | --- | --- | --- |
| | | | $\gamma_{FL} = 0.2$ | $\gamma_{FL} = 1$ | $\gamma_{FL} = 5$ | | |
| MSPE | 0, 2 | 93.4(1.7) | 85.0(2.0) | 131.3(1.7) | 182.5(1.9) | 138.2(0.6) | 136.2(0.6) |
| | 0, 4 | 86.3(1.6) | 55.8(1.4) | 105.8(1.6) | 165.3(1.4) | 133.6(0.6) | 132.3(0.6) |
| | 3, 2 | 277.8(5.3) | 340.6(12.9) | 353.6(10.6) | 674.0(13.1) | 532.3(8.5) | 483.2(6.0) |
| | 3, 4 | 163.9(2.2) | 115.7(3.6) | 128.5(4.2) | 221.9(7.3) | 335.0(4.8) | 213.4(2.7) |
| RI | 0, 2 | 0.72(0.009) | 0.47(0.012) | 0.66(0.004) | 0.66(0.002) | 0.28(0.000) | 0.29(0.000) |
| | 0, 4 | 0.72(0.010) | 0.46(0.007) | 0.69(0.004) | 0.66(0.002) | 0.28(0.000) | 0.29(0.000) |
| | 3, 2 | 0.79(0.004) | 0.58(0.012) | 0.78(0.007) | 0.65(0.006) | 0.28(0.000) | 0.29(0.000) |
| | 3, 4 | 0.80(0.003) | 0.57(0.011) | 0.82(0.007) | 0.85(0.003) | 0.28(0.000) | 0.29(0.000) |

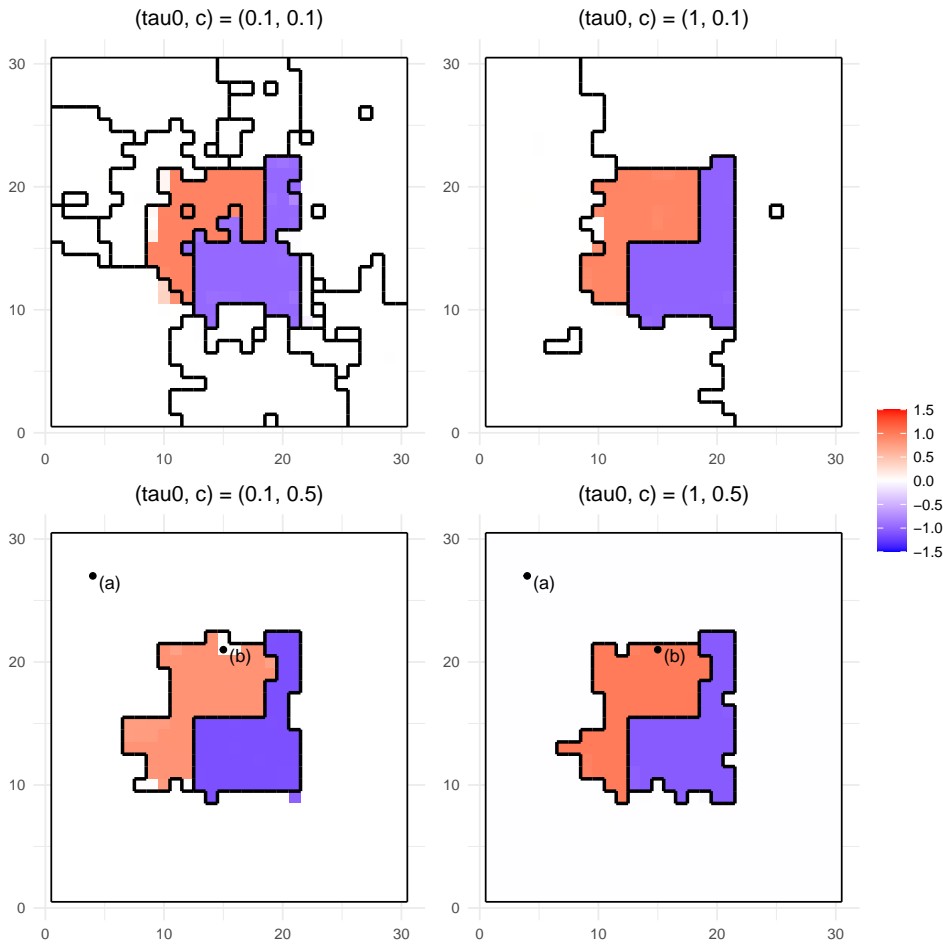

Figure A1: T-LoHo posterior median estimate of $\boldsymbol{\beta}$ when $(\vartheta,\text{SNR}) = (0,4)$ with different choice of hyperparameters $(\tau_0, c) \in \{(0.1, 0.1), (1, 0.1), (0.1, 0.5), (1, 0.5)\}$. Black boundaries illustrates the point estimate of cluster obtained from MCMC samples, based on Dahl's method Dahl (2006).

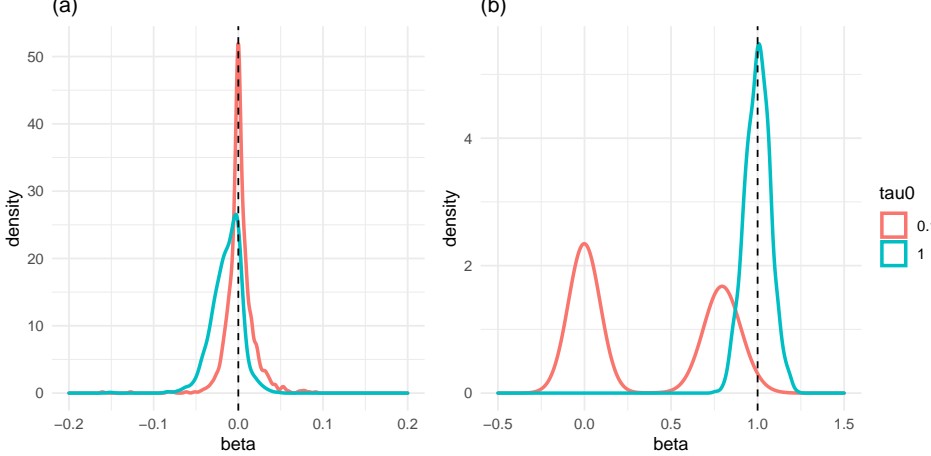

Figure A2: Posterior distribution of $\beta_j$ at two different locations, (a) and (b) as displayed in bottom panel of fig.A1. Red line is when $\tau_0 = 0.1$, blue is when $\tau_0 = 1$. Dashed line denotes the true value.

**Selection and sensitivity analysis of $\tau_0$**  Hyperparameter $\tau_0 > 0$ controls the scale of the global shrinkage parameter, $\tau \sim C^+(0, \tau_0)$. As $\tau_0$ approaches to 0, posterior distribution of $\boldsymbol{\beta}$ more concentrates at 0 for zero coefficients as we can see in the left panel of fig.A2. However at the same time, a too small $\tau_0$ can introduce a stronger bias towards zero for nonzero signals as shown in the right panel of fig.A2 (although this bias may disappear as the magnitude of signal increases). Thus, there is a trade-off between shrinkage and bias when selecting $\tau_0$. In practice, if $\boldsymbol{\beta}$ is assumed to be very sparse (i.e. contains many zeros) then a small $\tau_0$ is preferred because the benefit of shrinking surpasses the loss due to bias. Under the formulation of T-LoHo, it is desirable to use a small $\tau_0$ if the clusterwise parameter $\tilde{\boldsymbol{\beta}}$ is assumed to be very sparse. Recently, Piironen and Vehtari (2017) proposed to select $\tau_0$ based on the prior guess for the number of relevant variables. Following their approach, we suggest $\tau_0 = p_0/(K_0 - p_0)$ where $p_0$ is a prior guess for the number of nonzero clusters, $K_0$ is a prior guess for the total number of clusters. In our simulation setting, $\tau_0^* = 2/(3 - 2) = 2$ would be an appropriate choice of $\tau_0$ if true is known.

According to table A1, when we have an independent design ($\vartheta = 0$), $\tau_0 = 1$ gives slightly better results in terms of MSPE compared to $\tau_0 = 0.1$. This is because the shrinkage effect is not well reflected in the posterior median estimate $\hat{\boldsymbol{\beta}}$, while the bias is more emphasized when $\tau_0$ gets smaller. However when we have a dependent design ($\vartheta = 3$), $\tau_0 = 0.1$ gives slightly better results in terms of MSPE compared to $\tau_0 = 1$. This may be explained by the fact that dependency of $\mathbf{X}$ leads to many falsely identified clusters, where the benefit of shrinking those is more emphasized.

**Selection and sensitivity analysis of $c$**  Hyperparameter $c$ controls the geometric decay rate in the prior of $K$, i.e., $\Pr(K = k) \propto (1 - c)^k$. As $c$ becomes closer to 1, it more penalizes on the cluster generation and gives fewer clusters. Assumption (A-4) of Theorem 1 requires $1 - c \geq p^{-c_\alpha}$, which suggests that one should not choose the value of $c$ too close to 1 unless the number of vertices $p$ is very large. Indeed, we notice that when $c = 0.9$, the MSPE increases according to table A1.

Table A1 also shows that Rand index sharply drops down when $c = 0.1$, and it is even more emphasized when $\tau_0 = 0.1$. Indeed, at the top panel of fig. A1 we can see that the cluster estimate is highly compartmentalized. This phenomenon can be characterized with our previous discussion in Section 3.2, where we describe clustering as a trade-off between the Bayes factor $\mathcal{L} = p(\boldsymbol{y}|\mathcal{M}_2)/p(\boldsymbol{y}|\mathcal{M}_1)$ and the the penalization prior $p(K) \propto (1 - c)^K$. As $\tau_0$ becomes close to 0, induced density $\pi_\Delta$ has a sharper spike around the origin, which makes the alternative $\delta \sim \pi_\Delta$ behave more similar to the null $\delta = 0$. As $c$ becomes too close to 0, the effect of penalization prior $p(K)$ fades out, which leads to many redundant cluster estimates. Therefore, we recommend not to choose $c$ too close to 0.

In summary, we suggest to use $c = 0.5$ as a default and not to use too small or too large $c$. If the number of vertices in $G$ is sufficiently large, Assumption (A-4) gives a more wider choice of $c$, and users may choose a different value of $c$ based on their prior belief on the model size or some model selection criteria such as the Watanabe-Akaike information criterion (WAIC; Watanabe, 2010).

### A5.2  Simulation Studies With More Clusters

We performed additional simulation studies when there are multiple clusters in the data. Two scenarios are considered: 1) four nonzero clusters with values $-2, -1, 1, 2$ respectively; 2) eight nonzero clusters with values $-3, -3, -2, -1, 1, 2, 3, 3$ respectively. See fig. A3 for the graphical illustration. Similar as before, we compared the MSPE and Rand index under the setting of $n = 250$ and $p = 900$ in tables A2 and A3. The T-LoHo hyperparameters are set as $(\tau_0, c) = (1, 0.5)$ and $\gamma_{FL} \in \{0.2, 1, 5\}$. When sample size $n$ becomes smaller, it will be more challenging to identify clusters when the number of clusters is larger. In theorem 1, $g_n^*$, the size of the spanning tree space compatible with nonzero clusters, increases as the number of nonzero clusters increases. Assumption (A-1) states that it should be upper bounded by $n/\log p$ asymptotically, which may not hold when $n$ is small. Indeed, we observed that T-LoHo, FL, GOSCAR, and BGL all suffer from a much poorer performance.

Table A2: Performance comparison based on average MSPE and RI over 100 replicated simulations. Standard error is given in parentheses and time is in sec. RI=1 indicates exact recovery of true cluster.

| $K=4$ | $\vartheta$ | SNR | T-LoHo | STGP | FL (0.2) | FL (1) | FL (5) |
|---|---|---|---|---|---|---|---|
| | 0 | 2 | **85.2(3.1)** | 190.7(1.9) | 167.3(4.4) | 332.6(8.4) | 485.2(4.7) |
| MSPE | 0 | 4 | **34.9(1.4)** | 172.3(1.4) | 91.7(2.8) | 197.7(7.2) | 365.0(3.8) |
| | 3 | 2 | **399.0(14.0)** | 441.8(8.4) | 566.7(21.8) | 656.4(23.9) | 1050.2(32.4) |
| | 3 | 4 | **206.5(7.1)** | 361.0(7.5) | 304.6(9.4) | 377.9(12.5) | 649.9(18.7) |
| | 0 | 2 | **0.91(0.007)** | 0.65(0.010) | 0.83(0.008) | 0.76(0.008) | 0.58(0.002) |
| RI | 0 | 4 | **0.95(0.005)** | 0.63(0.008) | 0.86(0.009) | 0.77(0.013) | 0.61(0.002) |
| | 3 | 2 | **0.88(0.004)** | 0.77(0.003) | 0.61(0.011) | 0.79(0.006) | 0.78(0.003) |
| | 3 | 4 | **0.92(0.003)** | 0.75(0.004) | 0.60(0.011) | 0.80(0.006) | 0.82(0.002) |

Table A3: Performance comparison based on average MSPE and RI over 100 replicated simulations. Standard error is given in parentheses and time is in sec. RI=1 indicates exact recovery of true cluster.

| $K=8$ | $\vartheta$, SNR | T-LoHo | STGP | FL (0.2) | FL (1) | FL (5) |
|---|---|---|---|---|---|---|
| | 0, 2 | 427.2(9.7) | **414.5(4.3)** | 658.0(20.1) | 1282.1(23.3) | 1574.7(13.9) |
| MSPE | 0, 4 | **251.8(6.2)** | 352.7(3.3) | 405.0(12.4) | 934.4(33.0) | 1271.0(11.2) |
| | 3, 2 | 2285.1(61.7) | **1424.9(28.8)** | 2258.7(71.7) | 2814.4(81.4) | 4747.6(146.3) |
| | 3, 4 | 1189.3(31.4) | **978.1(20.1)** | 1364.8(47.2) | 1723.1(57.1) | 2944.4(87.3) |
| | 0, 2 | **0.76(0.006)** | 0.59(0.006) | 0.72(0.006) | 0.54(0.008) | 0.57(0.004) |
| RI | 0, 4 | **0.84(0.006)** | 0.61(0.005) | 0.78(0.005) | 0.64(0.009) | 0.61(0.002) |
| | 3, 2 | **0.71(0.006)** | 0.62(0.005) | 0.67(0.006) | 0.74(0.003) | 0.67(0.004) |
| | 3, 4 | **0.78(0.006)** | 0.64(0.004) | 0.68(0.005) | 0.77(0.004) | 0.73(0.003) |

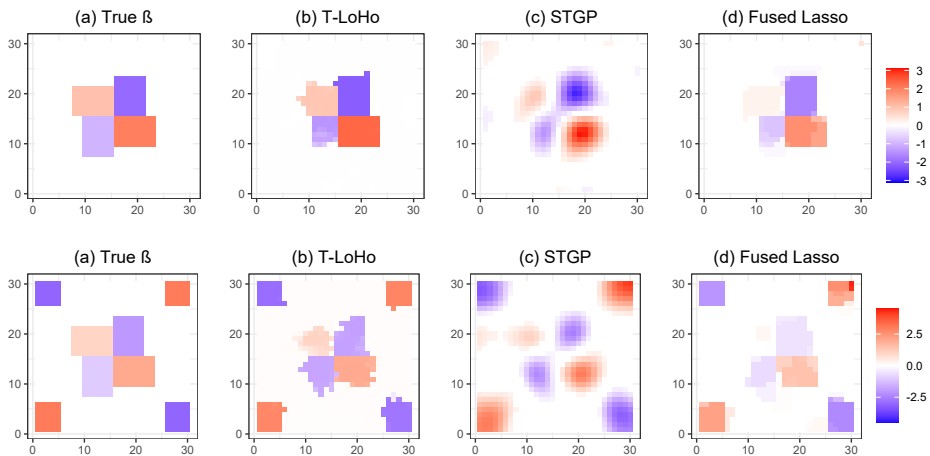

Figure A3: True and fitted results with four (first row), eight (second row) nonzero clusters. Here $(\vartheta, \text{SNR}) = (0, 4)$. (a) True coefficient image $\boldsymbol{\beta}$; (b) T-LoHo estimate with $(\tau_0, c) = (1, 0.5)$; (c) STGP estimate after thresholding; (d) FL estimate with $\gamma_{FL} = 0.2$.

The performance of all methods deteriorates as the number of nonzero clusters increases. The results are not surprising because, for many high-dimensional regularization methods, it is known that the learning rate often depends on the size of the true model (the number of clusters in our case) relative to the sample size. Except in the case when $K = 8$ with $\vartheta = 3$, T-LoHo still outperforms other baselines in terms of both MSPE and Rand index. It will be of interest to improve the performance of T-LoHo when predictors are correlated, sparsity level is relatively low, and it has many nonzero clusters. We left these as a future working direction.