# OpenReview forum: "T-LoHo: A Bayesian Regularization Model for Structured Sparsity and Smoothness on Graphs"
_NeurIPS.cc/2021/Conference — NeurIPS 2021 Poster_

### Official Review · Reviewer_M8yM · 2021-07-09

**Rating:** 7
**Confidence:** 3

**Summary:**

The paper proposes a new prior for high dimensional parameters with graphical relations. It can be regarded as an extension of univariate horseshoe shrinkage prior to multivariate setting by borrowing ideas from studies on homogenous sparse priors and tree-based graph partition prior. The paper describes posterior sampling strategy with efficient implementation details and theoretical guarantees. In the end, the paper demonstrates its methods via both simulated and real data experiments.

**Limitations And Societal Impact:**

Yes, the authors adequately addressed the potential negative societal impact of their work.

**Main Review:**

The paper extends the widely used univariate horseshoe prior to multivariate case. The task itself is not new and some of the ideas have their roots in previous literatures, however, the paper combines them in a non-trivial way. By introducing the low-rank structure for prior distribution, the paper naturally incorporates cluster information into prior. Then the paper uses the random minimum spanning tree approach introduced by [1] to model the partition prior. The remaining computation and theoretical proof techniques are borrowed from previous literatures. But the way they are combined are novel.

The paper is technically sound. The methods are clearly stated, and details of efficient implementation are included. The theoretical results are rigorously proved. In addition, the author uses simulation studies to demonstrate the benefits of their proposed methods compared with other widely used approaches. The simulation results are adequately interpreted. Besides, the author(s) also demonstrate(s) the method via real data application.

The paper is clearly written and well-written. Many informative figures are much appreciated. The details of method and experiments are described clearly. With that being said, the section 3.2 is a bit hard to follow. I think it will be better if the author(s) can give more high-level intuitive explanation.

Given the task of the paper is widely attractive, the Bayesian approach offered by the paper will be interesting to many researchers and practitioners. The method given by the paper is new and the benefit of it is shown by well-designed experiments. With that being said, most of results and techniques seem to be borrowed from somewhere else, and thus may not be surprising.

[1] Zhao Tang Luo, Huiyan Sang, and Bani Mallick. A Bayesian contiguous partitioning method for learning clustered latent variables. Journal ofMachine Learning Research, 22(37):1–52, 2021.


**Time Spent Reviewing:**

4

---

> ### Author Response · Authors · 2021-08-09
> **Author Response to Reviewer M8yM**
>
> We thank the reviewer for the accurate and nice summary of our work. As the reviewer pointed out, we extend the random spanning tree partition approach to a random spanning tree forest partition and embed it in a low-rank horseshoe framework. We remark that T-LoHo is not merely a simple combination where the horseshoe shrinkage prior and partition prior in T-LoHo handle sparsity and clustering separately. Instead, the way we combine them enables a very interesting complementary effect. On one hand, our theoretical investigation based on Bayes factor reveals that the horseshoe shrinkage component of T-LoHo plays a profound role in clustering by facilitating cluster fusion when signal difference is low and improving cluster identification when signal difference is high. We will revise Section 3.2 to further highlight this finding and move some too technical materials to the Appendix. On the other hand, the partition model allows us to account for the clustered dependence of local shrinkage parameters to facilitate sparsity learning.

---

### Official Review · Reviewer_ESxP · 2021-07-11

**Rating:** 6
**Confidence:** 2

**Summary:**

The authors propose a multi-dimensional horseshoe prior for Bayesian inference that enforces clustering of the (non)zero parameters. The main contribution of the paper is to make this approach more flexible, feasible and computationally efficient.

**Limitations And Societal Impact:**

In the paper, I could not find any comment about the potential negative societal impact of the results. On the other hand, I can hardly think about social issues related to this work, except for possible unfair effects on the obtained models coming from a naive graph-based selection process of the model parameters.

**Main Review:**

strengths:
Imposing structured sparsity can have many practical applications and making this computationally feasible may have a great impact on the community. The inclusion of the scheme in a Bayesian inference setup has many advantages from the theoretical perspective. The paper is technically sound and the proposed method comes with posterior consistency guarantees.

weaknesses:
The possibly unconvincing side of the paper is the value of its net contribution compared with the state-of-the-art. It is not clear if similar graph-structured priors have never been used in Bayesian learning and if their use to enforce parameter structured smoothness is new. A technical novelty of the paper is to go beyond the piecewise constant assumption of existing shrinkage approaches. The authors could have explained more explicitly what is the technical challenge of this extension. From what the authors write in Section 2, it looks like the key contribution relies on the inclusion of a new covariance matrix in the horseshoe prior (as the partition approximation scheme seems to be taken from previous work). The authors should explain better how this leads to the claimed computational advantages.

questions:
- One of the main motivations of this work is the fact that "fixed chain or tree orders may not be compatible with the true order with respect to its context G". How is the order-within-a-context defined?
- Are there examples of feasible or quasi-feasible regularization schemes that take into account the whole graph structure?
- How are the theoretical results listed at the end of Section 1, e.g. posterior consistency, different from standard guarantees that can be obtained with other Bayesian methods?
- What is a practical situation where a multi-dimensional parameter needs to show some sort of graph-neighbour continuity?
- The proposed covariance matrix is claimed to capture at the same time fusion and sparsity. Is this obtained by including a tunable diagonal factor?
- The goal of the new graph-clustering prior is to allow non-connected components in the graph. How does this come from the proposed form of the covariance matrix? And why is this important from an applications point of view?

**Time Spent Reviewing:**

3 hours

---

> ### Author Response · Authors · 2021-08-09
> **Author Response to Reviewer ESxP**
>
> Thank you for taking the time to give valuable feedback and insightful comments.
>
> Shrinkage priors utilizing graph structures have been developed in Bayesian learning. The main body of previous work is to develop a Bayesian version of fused learning where a univariate shrinkage prior is assigned to the edge differences. We cited Bayesian fused lasso [Kyung10], NEG prior[Shimamura19], Bayesian $t$ shrinkage [Song20] and Bayesian horseshoe fused shrinkage [Banerjee21] as examples. However, building fused shrinkage on the whole graph may not be the most statistically efficient way for clustering tasks because a large number of edges have to be identified and removed from the total edge set of $G$. Indeed, the computation of such methods are often quite expensive for a general graph with many edges. To address the computation, a number of work considered building a model on a *fixed* chain graph such as a time-ordered chain [Song20] and Depth First Search ordered chain [Padilla18]. However, using a fixed chain to explain the piecewise constant signal on a graph is not suitable for the clustering tasks, especially when the graph is dense, because the true partition of a graph may not be recovered by removing edges from a predetermined chain. In contrast, to obtain the same partition, an adaptive spanning forest method only requires identifying a much smaller edge set to remove from the tree edge set. We are the first to propose a graph-structured sparsity model based on the adaptive spanning forest prior. The methodology, theory, and computation algorithms are a novel addition to the current literature.
>
> The proposed low-rank covariance matrix has several benefits:
> - It maintains interpretability of $\beta$, a key distinction from the Bayesian Compressed Regression model [Guhaniyogi15]
> - It gives computational benefits by exploiting the low-rank structure and embracing the continuous shrinkage prior. Spike-and-slab type priors are prohibitive here due to the combinatorial complexity of updating the discrete indicators and infeasibility of block updating of model parameters [Bhadra19]. Plus, the orthonormal rows of $\Phi$ balances the effect of local shrinkage parameters across the clusters with varying sizes.
> - It provides a very interesting complementary effect on clustering. Our theoretical investigation based on Bayes factor reveals that the horseshoe shrinkage component of T-LoHo plays a profound role in clustering by facilitating cluster fusion when signal difference is low and improving cluster identification when signal difference is high.
>
> We next address your questions:
> > One of the main motivations of this work is the fact that "fixed chain or tree orders may not be compatible with the true order with respect to its context G". How is the order-within-a-context defined?
>
> We're sorry for the confusing wording, the 'true order' in that sentence (line 41) should be replaced by 'true cluster'. The forest $\mathcal{F}$ is said to be *compatible* with graph partition $\Pi$ if we can construct $\Pi$ by cutting some of its edges(line 135).
>
> > Are there examples of feasible or quasi-feasible regularization schemes that take into account the whole graph structure?
>
> Fused Lasso on graph [Tibshirani11] and graph OSCAR [Yang12] both take the whole graph structure into account when regularizing the parameters. Efficient implementations such as dual path algorithm and ADMM have been developed which makes the model work for the large-sized graphs. Another different but relevant approach is the Bayesian graph Laplacian [Chakraborty19]. It utilizes EM algorithm to learn a graph structure using prior precision matrix $\Lambda$, where the resulting cluster does not guarantee to respect any contiguity constraint regarding the context $G$ of the problem. In addition to the two baselines in the original submission, we have added additional competing model results including graph OSCAR(GOSCAR) and Bayesian graph Laplacian(BGL). The setting is same as section 4.1 with $\vartheta =0$ and $SNR=4$.
>
> |        | T-LoHo | STGP  | Fused Lasso | GOSCAR | BGL |
> | :--- :       |    :----:   |    :---: | :---: |:---: |:---: |
> | MSPE      |  **24.39(19.57)**  | 86.27(15.75) | 55.79(14.22) | 144.32(6.51) | 132.31(5.69) |
> | Rand Index   | **0.95(0.05)** | 0.72(0.10)  | 0.79(0.03) | 0.72(0.002) | 0.66(0.004) |
>
> The results indicate that T-LoHo model achieves better prediction and cluster estimation compared to other regularization schemes that take into account the whole graph structure. Although T-LoHo model does not explicitly regularize the whole graph structure, it achieves better performance by exploring the spanning forest space using the adaptive spanning forest prior.
>
> > How are the theoretical results listed at the end of Section 1, e.g.posterior consistency, different from standard guarantees that can be obtained with other Bayesian methods?
>
> The theoretical results of our method are built upon a seminal paper by [Ghosal07], which laid out the general posterior concentration theory and three main conditions, namely, parameter space complexity, prior concentration, and negligibility of sieve complement in terms of prior mass. However, it is known that for specific high dimensional data and prior models, the verification of these conditions is usually quite challenging and needs to be treated case by case with care. Although our regression data model has been considered in previous theoretical work [Song17], the proposed T-LoHo prior is new and requires new ways to construct the sieves, to check the prior concentration probabilities, and to check parameter space complexity that involve nontrivial graph and spanning tree conditions and theories.
>
> > What is a practical situation where a multi-dimensional parameter needs to show some sort of graph-neighbour continuity?
>
> T-LoHo is a very general prior for the selection and clustering of a multi-dimensional parameter with graphical relations. There are many examples and potential applications of this method.  As we showed in the real data, trend filtering on (road) graph is one such example where true signals often respect contiguity constraint on graphs. Scalar on image [Kang18] or image on scalar regression [Chen16] with a sparse and clustered multi-dimensional parameter have also gained great popularity in many biomedical imaging studies to find imaging biomarkers. T-LoHo can also be applied to network data such as gene networks to find a group of important genes connected by a biological network to differentiate diseased and non-diseased individuals.
>
> > The proposed covariance matrix is claimed to capture at the same time fusion and sparsity. Is this obtained by including a tunable diagonal factor?
>
> The diagonal factor of the covariance matrix $\Lambda$ captures the sparsity, by imposing half-Cauchy prior on the diagonal elements $\lambda_1,\ldots,\lambda_K$. Fusion is captured by the projection matrix $\Phi$, built upon the graph partition $\Pi$. Diagonal factor $\Lambda$ is updated at Step 3 in Algorithm 1, with details provided in Appendix A1.
>
> > The goal of the new graph-clustering prior is to allow non-connected components in the graph. How does this come from the proposed form of the covariance matrix? And why is this important from an applications point of view?
>
> While the primary goal of T-LoHo model is to make inference on the sparse and clustered parameters, allowing to incorporate prior knowledge on non-connected components of the graph is important from both theoretical and practical points of view. We provided posterior contraction result with a general graph $G$ with the number of initial connected components $n_c$. Assumption (A-1) and the proof in Appendix A4.2 give an insight on the role of $n_c$ by limiting its growth rate. From the practical point of view, a graph with multiple initial connected components frequently arises in many fields such as road networks and gene networks. Also, allowing a non-connected graph is useful when one wants to incorporate unstructured predictors(e.g. age, sex, weight) with graph-structured predictors(e.g. gene expression level) together. In this case, one can construct a graph that has multiple components by letting individual unstructured predictors as isolated vertices, similar to the left panel of fig. 1.
>
> **References**
>
> Banerjee, S. (2021) Horseshoe shrinkage methods for Bayesian fusion estimation. arXiv.
>
> Bhadra, A., Datta, J., Polson, N. G., & Willard, B. (2019) Lasso meets horseshoe: A survey. Stat. Sci.
>
> Chakraborty, S., & Lozano, A. C. (2019) A graph Laplacian prior for Bayesian variable selection and grouping. Comput. Stat. Data Anal.
>
> Chen, Y., Wang, X., Kong, L., & Zhu, H. (2016) Local region sparse learning for image-on-scalar regression. arXiv
>
> Guhaniyogi, R., & Dunson, D. B. (2015) Bayesian compressed regression. JASA.
>
> Ghosal, S., & Van Der Vaart, A. (2007) Convergence rates of posterior distributions for noniid observations. Ann. Stat.
>
> Kang, J., Reich, B. J., & Staicu, A. M. (2018) Scalar-on-image regression via the soft-thresholded Gaussian process. Biometrika
>
> Kyung, M., Gill, J., Ghosh, M., & Casella, G. (2010) Penalized Regression, Standard Errors, and Bayesian Lassos. Bayesian Anal.
>
> Padilla, O. H. M., Sharpnack, J., Scott, J. G., & Tibshirani, R. J. (2018) The DFS Fused Lasso: Linear-Time Denoising over General Graphs. J. Mach. Learn. Res.
>
> Shimamura, K., Ueki, M., Kawano, S., & Konishi, S. (2019) Bayesian generalized fused lasso modeling via NEG distribution. Commun. Stat. Theory Methods
>
> Song, Q., & Liang, F. (2017) Nearly optimal Bayesian shrinkage for high dimensional regression. arXiv
>
> Song, Q., & Cheng, G. (2020) Bayesian fusion estimation via t shrinkage. Sankhya A
>
> Tibshirani, R. J., & Taylor, J. (2011) The solution path of the generalized lasso. Ann. Stat.
>
> Yang, S., Yuan, L., Lai, Y. C., Shen, X., Wonka, P., & Ye, J. (2012) Feature grouping and selection over an undirected graph. KDD'12

---

> > ### Comment · Reviewer_ESxP · 2021-09-10
> > **thank you for your answers**
> >
> > After reading the authors' response I would like to confirm my score.

---

### Official Review · Reviewer_M78Z · 2021-07-16

**Rating:** 6
**Confidence:** 1

**Summary:**

This paper propose a new prior referred to as Tree-based Low-rank Horseshoe (T-Loho), which can incorporate relationship among variables as a graph.
Theoretical characteristics of the clustering effects and posterior concentrations are investigated.
In computational experiments based on both synthetic and real-world data, T-Loho turned out to be superior to fused lasso and soft-thresholded Gaussian process in terms of both regression performance and support recovery.

**Limitations And Societal Impact:**

In computational experiments, comparison to Bayesian fused lasso is missing.
More experiments demonstrating the power of tree-based graph partition prior are expected.

**Main Review:**

The paper is presented with in-depth technical details, and therefore not easy to read from time to time.

**Time Spent Reviewing:**

4

---

> ### Author Response · Authors · 2021-08-09
> **Author Response to Reviewer M78Z**
>
> We thank the reviewer for taking time to review our paper.
>
> We agree with the reviewer that the current manuscript involves many technical details, especially on the theories of clustering effects and posterior contraction results. But we feel these theoretical results can provide valuable insights on the shrinkage/clustering properties of the T-LoHo prior and useful guidance in terms of prior hyperparameter selections. For example, it appears on the surface that the horseshoe shrinkage prior and partition prior in T-LoHo handle sparsity and homogeneity separately, but our theoretical investigation based on Bayes factor reveals that shrinkage component in T-LoHo can facilitate cluster fusion when signal is low and improving cluster identification when signal is high. Another example is the choice of hyperparameter $c$ that controls the strength of penalization on large numbers of clusters. Our theory suggests that one should not choose the value of $c$ too close to 1 unless the number of vertices $p$ is very large. Indeed, the sensitivity analysis with respect to $c$ in Appendix A5.2 agrees well with these theoretical findings. In the revised manuscript, we will highlight more on these high-level findings and leave some of the technical proofs to Appendix.
>
> We also thank the reviewer for suggesting more baseline methods for comparisons. Bayesian fused lasso [Kyung10] is certainly relevant literature. But since the posterior mode of Bayesian fused lasso is equivalent to the frequentist fused lasso method which we already included, we anticipate that these two competing methods will have very similar performances. In addition to the two baselines in the original submission, we have added additional competing model results including graph OSCAR [Yang12] and Bayesian graph Laplacian [Chakraborty19] to demonstrate the power of T-LoHo model. OSCAR[Bondell08] and graph OSCAR[Yang12] models are based on penalized least squares with octagon-shaped penalty function that shrinks some coefficients to exactly zero and forms clusters of coefficient. Bayesian graph Laplacian[Liu14, Chakraborty19] shares the same goal with T-LoHo, that is, finding sparse and clustered parameters, but the cluster they considered does not depend on the context $G$ given as a graph. The precision matrix $\Lambda$ of the graph laplacian prior can be decomposed as $D-S$, which has a very similar form as a graph Laplacian matrix of a graph with degree $D$ and weight adjacency matrix $S$.
> The way the graph Laplacian model works is to learn a graph structure using prior precision matrix $\Lambda$, where the resulting cluster does not guarantee to respect any contiguity constraint regarding the context of the problem. Nevertheless, we have included this model as a new baseline. The setting is same as section 4.1 with $\vartheta =0$ and $SNR=4$.
>
> |        | T-LoHo | STGP  | Fused Lasso | GOSCAR | BGL |
> | :--- :       |    :----:   |    :---: | :---: |:---: |:---: |
> | MSPE      |  **24.39(19.57)**  | 86.27(15.75) | 55.79(14.22) | 144.32(6.51) | 132.31(5.69) |
> | Rand Index   | **0.95(0.05)** | 0.72(0.10)  | 0.79(0.03) | 0.72(0.002) | 0.66(0.004) |
>
> The results indicate that T-LoHo model achieves better prediction and cluster estimation compared to other regularization schemes that take into account the whole graph structure.
>
> **References**
>
> Bondell, H. D., & Reich, B. J. (2008). Simultaneous regression shrinkage, variable selection, and supervised clustering of predictors with OSCAR. Biometrics, 64(1), 115-123.
>
> Chakraborty, S., & Lozano, A. C. (2019). A graph Laplacian prior for Bayesian variable selection and grouping. Computational Statistics & Data Analysis, 136, 72-91.
>
> Kyung, M., Gill, J., Ghosh, M., & Casella, G. (2010). Penalized Regression, Standard Errors, and Bayesian Lassos. Bayesian Analysis, 5(2), 369-412.
>
> Liu, F., Chakraborty, S., Li, F., Liu, Y., & Lozano, A. C. (2014). Bayesian regularization via graph Laplacian. Bayesian Analysis, 9(2), 449-474.
>
> Yang, S., Yuan, L., Lai, Y. C., Shen, X., Wonka, P., & Ye, J. (2012, August). Feature grouping and selection over an undirected graph. In Proceedings of the 18th ACM SIGKDD international conference on Knowledge discovery and data mining (pp. 922-930).

---

### Official Review · Reviewer_A5dn · 2021-07-18

**Rating:** 7
**Confidence:** 4

**Summary:**

The authors generalize the univariate Bayesian horseshoe shrinkage prior to the multivariate setting and propose a new prior that detects sparsity and encourages smoothness within clusters of variables. The paper illustrates the application of the prior in Bayesian regression and present an MCMC algorithm. Numerical experiments with synthetic data and results with one real dataset illustrate the potential advantages over alternative methods.

**Main Review:**

The paper is well-written. The proposed prior is novel and the authors develop an efficient reversible jump MCMC sampler. The discussion on clustering effects provides valuable insights into the nature of the proposed prior and the authors provide a posterior contraction result.
The weaker parts of the paper are the discussion of related work and the simulation experiments.

(1) The proposed method is compared against only two baselines, with one dating from 2005. Although fused lasso should definitely be included as a classical baseline, I think there are many alternatives for problems of this kind. Are the authors suggesting that there has been little interest in the analyzed problem and that there has only been one suitable competitive method proposed in the past decade? I would think that network lasso [R1] and its derivatives, the graph-based regularization of [R2], OWL and OSCAR regularization, and Bayesian graph Laplacian regularization [R3] would be candidate alternative methods.

(2) The synthetic experiments only explore a scenario of two clusters. The problem would seem to be much more complex when there are multiple clusters in the data. The algorithm appears to perform well for the real data-set (where it appears to identify more clusters). Results for synthetic experiments with more clusters would provide insight into the behaviour – it’s not clear if many more iterations are required for a satisfactory exploration.

[R1] Hallac, David, Jure Leskovec, and Stephen Boyd. "Network lasso: Clustering and optimization in large graphs." Proceedings of the 21th ACM SIGKDD international conference on knowledge discovery and data mining. 2015.

[R2] Li, Yuan, et al. "Graph-Based Regularization for Regression Problems with Alignment and Highly Correlated Designs." SIAM J. Mathematics of Data Science 2.2 (2020): 480-504.

[R3] Liu, Fei, et al. "Bayesian regularization via graph Laplacian." Bayesian Analysis 9.2 (2014): 449-474.

** AFTER AUTHOR RESPONSE

My main concern with the paper was the inadequate comparison to other methods for addressing the same problem. In their response, the authors have indicated that they will include comparisons to several more recent baseline methods, and they have included results for preliminary experiments, which indicate that the proposed method provides a meaningful improvement. If the paper is revised to include more extensive experimental comparisons (which constitutes a relatively minor change), then I would consider it a valuable technical contribution. I have modified my recommendation to "good paper, accept"


**Time Spent Reviewing:**

5 hours

---

> ### Author Response · Authors · 2021-08-09
> **Author Response to Reviewer A5dn**
>
> We thank the reviewer for taking the time to give valuable feedback and insightful comments.
>
> We chose graphical fused lasso [Tibshirani11] as a baseline due to its popularity and availability of well-tested packages. There has been a continued interest in modeling sparse and smooth signals on a graph, starting from the fused lasso model on a chain graph [Tibshirani05]. We thank the reviewer for pointing out more relevant literature. We have taken the suggestion and conducted additional simulations using graph OSCAR [Yang12] and Bayesian graph Laplacian [Chakraborty19] under the same setting as section 4.1 with $\vartheta=0$ and $SNR=4$. As shown in the below table, T-LoHo still outperforms compared to others. The results will be added into the revised manuscript.
>
> |   | T-LoHo | STGP  | Fused Lasso | GOSCAR | BGL |
> | :--- :       |    :----:   |    :---: | :---: |:---: |:---: |
> | MSPE      |  **24.39(19.57)**  | 86.27(15.75) | 55.79(14.22) | 144.32(6.51) | 132.31(5.69) |
> | Rand Index   | **0.95(0.05)** | 0.72(0.10)  | 0.79(0.03) | 0.72(0.002) | 0.66(0.004) |
>
> Below, we summarize the connections and differences of T-LoHo with the two additional baselines and other relevant subsequent work of fused lasso. We will revise the Introduction section accordingly.
>
> * **Generalization of the fused lasso**.  Fused lasso penalty on chain graphs can be naturally extended to the edge difference penalty on a general graph $G$ [Tibshirani11]. It has been further generalized to the trend filtering on graphs [Wang16] by introducing graph difference operators. Also, there has been a recent advancement in theory [Liu13, Lin17] and computation [Arnold16, Padilla18] of the fused lasso model, to name a few. Network lasso [Hallac15] is a multivariate extension of the fused lasso, where parameter $\beta_j$ associated with node $j$ is now a $d$-dimensional vector. When $d=1$, it reduces to the fused lasso model that was included as a baseline in the original manuscript.
> * **Model with different penalty structure or prior**.  OSCAR[Bondell08] and OWL[Bogdan15] are definitely related literatures using different penalties on the parameters. [Yang12] proposed the graph OSCAR (which can be considered as a special case of graph OWL) and is added as a new baseline.
>     Bayesian fused lasso [Kyung10] is another relevant literature. But since the posterior mode of Bayesian fused lasso is equivalent to the frequentist fused lasso method which we already included, we anticipate that these two competing methods will have very similar performances.
>     Recently Bayesian t shrinkage [Song20] and horseshoe fused shrinkage [Banerjee21] has been proposed, but as we addressed in our manuscript, using a fixed chain to explain the piecewise constant signal on a graph is not suitable for clustering tasks, especially when the graph is dense, because the true partition of a graph may not be recovered by removing edges from a predetermined chain. We excluded those in our simulation studies because they only considered normal means model $(X=I)$.
> * **Different assumption on the parameters or predictors**.
>     * [Li20] considered the situation in which a 'covariance graph' governs both 1) the similarity among regression coefficients and 2) correlations among the covariates so that the design matrix $X$ is highly correlated. We emphasize that T-LoHo only imposes the former assumption(homogeneity), but not requiring the latter assumption (correlation of $X$ matches with the graph).
> [Li20] used the estimated covariance matrix $\hat{\Sigma}$ to generate the 'covariance graph' and regularize the parameters. Such estimate of $\hat{\Sigma}$ can be obtained by design matrix $X$ or side information such as pre-known cluster structure of the parameters.
> Thus, [Li20]'s model cannot cluster any parameters of $\beta$ in the case with independent design matrix $X$, since it can only generate an edgeless covariance graph.
>     * Bayesian graph Laplacian[Liu14, Chakraborty19] shares the same goal with T-LoHo, that is, finding sparse and clustered parameters, but the cluster they considered does not depend on the context $G$ given as a graph. The precision matrix $\Lambda$ of the graph laplacian prior can be decomposed as $D-S$, which has a very similar form as a graph Laplacian matrix of a graph with degree $D$ and weight adjacency matrix $S$.
> The way the graph Laplacian model works is to learn a graph structure using prior precision matrix $\Lambda$, where the resulting cluster does not guarantee to respect any contiguity constraint regarding the context of the problem. Nevertheless, we have included this model as a new baseline.
>
>
> Regarding the reviewer's second question about the effect of the number of true clusters on model performance,  we also conducted additional simulation studies when there are multiple clusters exist in the parameter. Two scenarios are considered: 1) four nonzero clusters with values -2,-1,1,2 respectively, 2) eight nonzero clusters with values -3,-3,-2,-1,1,2,3,3 respectively. We compared the MSPE and Rand index under the setting of $n=250$, $p=900$, $\vartheta=0$ and $SNR=4$. T-LoHo hyperparameters are set by $(\tau_0,c)=(1, 0.5)$, and we collected 4,000 posterior samples with 10 thin-in rate and $10^4$ burn-in iterations.
>
> |   Four nonzero clusters     | T-LoHo | STGP  | Fused Lasso |
> | :--- :       |    :----:   |    :---: | :---: |
> | MSPE      |  **34.92(14.24)**  | 172.27(14.21) | 91.70(28.05) |
> | Rand Index   | **0.95(0.04)** | 0.63(0.08)  | 0.87(0.02) |
>
> |   Eight nonzero clusters     | T-LoHo | STGP  | Fused Lasso |
> | :--- :       |    :----:   |    :---: | :---: |
> | MSPE      |  **251.81(61.85)**  | 352.74(32.67) | 405.04(124.00) |
> | Rand Index   | **0.84(0.06)** | 0.61(0.05)  | 0.77(0.04) |
>
> As shown in the table above, the performance of all methods deteriorates when there are more clusters as the reviewer conjectured, but T-LoHo still outperforms other baselines. The results are not surprising because, for many high dimensional regularization methods, it is known that the learning rate often depends on the size of the true model (the number of clusters in our case) relative to the sample size. The results and figure illustrations will be added to the revised manuscript.
>
> In addition to the comparison on the prediction and cluster estimation performance, trace plot analysis of T-LoHo model shows that roughly 20000-30000 iterations are needed for MCMC to converge when there are 8 clusters, whereas roughly 10000-20000 iterations are needed when there are 4 clusters, indicating that more iterations might be needed when the number of nonzero clusters increases.
>
> When sample size $n$ becomes smaller, it will be more challenging to identify clusters when the number of clusters is larger. Assumption (A-1) in our theorem states that the size of the spanning tree space compatible with nonzero clusters, which increases as the number of nonzero clusters increases, should be upper bounded by $n/(\log p)$ asymptotically. Indeed, we also experimented with the setting when $n=100$ with 8 number of clusters and observed that T-LoHo, fused lasso, graph OSCAR, and Bayesian graph Laplacian all suffer from a much poorer performance.
>
> We will revise our paper to address the reviewer's concerns and our manuscript will greatly benefit from the reviewer's comments.
>
> **References**
>
> Arnold, T. B., & Tibshirani, R. J. (2016). Efficient implementations of the generalized lasso dual path algorithm. Journal of Computational and Graphical Statistics.
>
> Banerjee, S. (2021). Horseshoe shrinkage methods for Bayesian fusion estimation. arXiv.
>
> Bogdan, M., Van Den Berg, E., Sabatti, C., Su, W., & Candès, E. J. (2015). SLOPE—adaptive variable selection via convex optimization. The Annals of Applied Statistics.
>
> Bondell, H. D., & Reich, B. J. (2008). Simultaneous regression shrinkage, variable selection, and supervised clustering of predictors with OSCAR. Biometrics.
>
> Chakraborty, S., & Lozano, A. C. (2019). A graph Laplacian prior for Bayesian variable selection and grouping. Computational Statistics & Data Analysis.
>
> Hallac, D., Leskovec, J., & Boyd, S. (2015, August). Network lasso: Clustering and optimization in large graphs. KDD'15.
>
> Kyung, M., Gill, J., Ghosh, M., & Casella, G. (2010). Penalized Regression, Standard Errors, and Bayesian Lassos. Bayesian Analysis
>
> Li, Y., Mark, B., Raskutti, G., Willett, R., Song, H., & Neiman, D. (2020). Graph-Based Regularization for Regression Problems with Alignment and Highly Correlated Designs. SIAM journal on mathematics of data science.
>
> Lin, K., Sharpnack, J. L., Rinaldo, A., & Tibshirani, R. J. (2017). A sharp error analysis for the fused lasso, with application to approximate changepoint screening. NIPS.
>
> Liu, J., Yuan, L., & Ye, J. (2013, May). Guaranteed sparse recovery under linear transformation. ICML.
>
> Liu, F., Chakraborty, S., Li, F., Liu, Y., & Lozano, A. C. (2014). Bayesian regularization via graph Laplacian. Bayesian Analysis.
>
> Padilla, O. H. M., Sharpnack, J., Scott, J. G., & Tibshirani, R. J. (2018). The DFS Fused Lasso: Linear-Time Denoising over General Graphs. J. Mach. Learn. Res.,
>
> Song, Q., & Cheng, G. (2020). Bayesian fusion estimation via t shrinkage. Sankhya A.
>
> Tibshirani, R., Saunders, M., Rosset, S., Zhu, J., & Knight, K. (2005). Sparsity and smoothness via the fused lasso. Journal of the Royal Statistical Society: Series B (Statistical Methodology).
>
> Tibshirani, R. J., & Taylor, J. (2011). The solution path of the generalized lasso. The annals of statistics.
>
> Wang, Y. X., Sharpnack, J., Smola, A. J., & Tibshirani, R. J. (2016). Trend Filtering on Graphs. Journal of Machine Learning Research.
>
> Yang, S., Yuan, L., Lai, Y. C., Shen, X., Wonka, P., & Ye, J. (2012). Feature grouping and selection over an undirected graph. KDD'12.

---

> > ### Comment · Reviewer_A5dn · 2021-08-25
> > **Thank you for the response**
> >
> > Thank you for the detailed response. The additional simulation results address my major concerns. I encourage the authors to include them in a revision. I think exploration of settings where there are more clusters constitutes an interesting and important research direction, but that can be left for a future paper.I have raised my recommendation to "good paper, accept".

---

> > > ### Author Response · Authors · 2021-08-26
> > > **Thanks!**
> > >
> > > We thank the review for taking the time to read our response. We will take your suggestions to revise our manuscript.

---

### Decision · Program_Chairs · 2021-09-27

**Decision:**

Accept (Poster)

**Comment:**

The reviewers all agree that the paper makes novel and valuable contributions. One the main concerns of the reviewers was that the paper was missing a sufficiently compelling discussion of how the method proposed in the paper compares with the state of the art, and an experimental comparison with more alternative methods. The responses to the different reviewers addresses well these concerns. The authors are strongly encouraged to include the new experimental results and the elements of discussion provided in the final version of the paper.